# Grass Meal Acts as a Probiotic in Chicken

Elena S. Bogdanova [1], Maria A. Danilova [1], Maria S. Smirnova [1], Elena V. Trubnikova [1,2,*], Hoa T. Nguyen [1], Svetlana N. Petrova [3], Sergey V. Apanasenok [4] and Alexei B. Shevelev [1]

1   DNA Methylome and Transcriptome Editing Laboratory, Vavilov Institute of General Genetics of Russian Academy of Sciences, Gubkina Str. 3, Moscow 119333, Russia; ledera@yandex.ru (E.S.B.); ms.maria.danilova@gmail.com (M.A.D.); mbarbotko@yandex.ru (M.S.S.); nguyenhoand001@gmail.com (H.T.N.); shevel_a@hotmail.com (A.B.S.)
2   Research Laboratory 'Genetics', Kursk State University, Radisheva Str. 33, Kursk 305000, Russia
3   I.I. Ivanov Kursk State Agrarian University, Karla Marksa Str. 70, Kursk 305021, Russia; svet-orl@yandex.ru
4   EdiVac LLC, Room 6 Oktyabrsky Prospekt 145, Lyubertsy 140000, Russia; 89191770721@mail.ru
*   Correspondence: tr_e@list.ru; Tel.: +7-910-311-1886

**Abstract:** Probiotics can act as an alternative to antibiotics in animal feeding, but their use is minimal due to their expensive production. Dry grass is rich with bacteria beneficial for animal feeding and can be used as a probiotic. However, data about the quantitative dependence of the grass microbiome on environmental factors and seasons remain insufficient for preparing "grass-meal-based probiotics". Four grass samples were collected in two geographically remote regions of Russia; their microbiome was characterized by metagenomic sequencing of 16S rDNA libraries and microbiological seeding, and biological testing of the grass meal was carried out on 6 groups of birds containing 20 Ross 308 cross broilers each for a period of 42 days. The positive control group (PC) obtained 16–25 mg/mL toltrazuril (coccidiostatic agent) and 0.5 mL/L liquid antibiotic enrostin (100 mg/mL ciprofloxacin and $10^6$ MU/mL colistin sulfate in the commercial preparation) within the drinking water, while the negative control group (NC) obtained no medicines. Four experimental groups were fed the diet supplemented with 1% grass meal over the period of 7–42 days of life; no commercial medicines were used here. A spontaneous infection with *Eimeria* was registered in the NC control groups, which caused the loss of 7 chickens. No losses were registered in the PC group or the two experimental groups. In two other experimental groups, losses of coccidiosis amounted to 10% and 15%, respectively. All specimens of the grass meal demonstrated a significant effect on the average body weight gain compared to NC. Taken together, these observations support the hypothesis that the grass meal may substitute toltrazuril for protecting the chickens from parasitic invasion and increase average daily weight gain (ADG) as effectively as the antibiotic enrostin.

**Keywords:** hay; grass meal; probiotic; chicken; coccidiosis; metagenomics; microbiome; phylloplane; *Eimeria tenella*

## 1. Introduction

Currently, poultry farming is the principal source of meat worldwide, providing the most available source of valuable protein [1]. Intensive development of four-line cross systems in chickens (e.g., Cobb 500 and Ross 308 fast-growing bred) and ameliorating cage, ventilation, climatic, feed distributing, and waste management facilities over the last 6 or 7 decades has resulted in feedstock conversion into muscle mass efficiency [2]. The feed conversion ratio (FCR) in these crosses attains 1.5–1.7 to 42 days of life [3,4].

Gross industrial farming of the broilers decreases manufacturing costs but makes the flock vulnerable to infection with *Campylobacter jejuni* [5], *Clostridium perfringens* [6], *Clostridium deficile* [7], several species of *Salmonella* [8], enteropathogenic *Escherichia coli* [9], *Eimeria* [10], avian leukosis virus [11], and *Enterococcus avium*, resulting in large economic losses to the poultry industry worldwide [12]. Antibiotics and antiparasitic medicines are commonly used at low doses for infectious disease prevention in broilers, thereby ameliorating their growth

and preventing losses. Nevertheless, the misuse and overuse of the drugs as growth promoters unavoidably lead to emerging antibiotic resistance in the broiler microbiota, including pathogens [13]. In 2015, the global annual consumption of antimicrobials per kg of animal product was estimated at 45 mg/kg, 148 mg/kg, and 172 mg/kg for cattle, chicken, and pigs, respectively [14]. Starting from this baseline, the global consumption of antimicrobials was expected to increase by 67%, from 63,151 ± 1560 tons to 105,596 ± 3605 tons between 2010 and 2030 [14]. The impact of antibiotic use for growth promotion in livestock and poultry production on the rise of antimicrobial resistance in bacteria led to the ban of this practice in the European Union in 2006 and a restriction of antimicrobial use in animal agriculture in Canada and the US [15]. A recently emerged paradigm of bioeconomy suggests using biological means of control for infection agents affecting poultry, including probiotics (live microbial preparations with antagonist activity toward pathogens), prebiotics, phytobiotics, bacteriophages, and their lysins [16,17].

Since 1973, probiotics have been suggested as an efficient and safe alternative to feeding antibiotics [2,18]. Traditionally, representatives of the genus *Lactobacillus* and taxonomically close groups (*Streptococci*, *Enterococci*) were used in this role [19]. As far as *Lactobacilli* are normal components of the chicken crop, small intestine, and cloaca, but not caeca, long-term survival of the administered bacteria was presumed. Therefore, strains with high resistance to acidic pH, bile, and pepsin with high adhesion ability to intestinal mucin were suggested to be most efficient, and special methods of selection for these traits were proposed [20]. Further, probiotics from other taxonomic lineages were successfully used. First, enterobacteria, including *E. coli* [21] and *Bacilli* [22], were used. There are reports about the high efficiency of *B. subtilis* strains isolated from chicken feces [23], despite the fact that the survival of *Bacillus* in the chicken gastric intestinal tract (GIT) seems doubtful. Later strains of *Clostridium* [24] and Ascomycetes yeast, e.g., *Saccharomyces boulardii* [25], were introduced to the practice. The most popular commercial probiotics available on the global market are Aviguard, Primalac, and Interbac made up of several species of *Lactobacillus* and *Bacillus* [26].

Significantly, probiotics, like antibiotics (e.g., enramycin and tylosin), confer resistance against *Eimeria* on the chicken, although they do not exhibit antagonism towards *Apikomplexa sporozoits* in vitro [27]. Moreover, the concept of the necessity of long-term persistence of the probiotics in GIT was revised. Obviously, the high efficiency of *Bacillus* strains' anti-pathogenic and growth-promoting effects on the chicken was acknowledged, although a pure aerobic metabolism does not allow *Bacilli* to vegetate under the chicken's GIT anaerobic conditions. Moreover, culture medium fermented with *Bacillus licheniformis* and *Bacillus subtilis* exerted a favorable impact on the GIT microbiota and average daily weight gain (ADG) of the broilers [2,28]. This confirms that the short-term influence of the probiotic-derived metabolite is sufficient for the favorable action of the overall probiotic. Therefore, the mechanism and final result of the anti-pathogenic action of the antibiotics and probiotics may be more similar than previously suggested.

Importantly, the impact of antibiotic preparations on the microbiome of caeca was described in [29]. The effect of the coccidiostat monensin and the growth promoters virginiamycin and tylosin on the caecal microbiome and metagenome of broiler chickens, 16S rRNA, and total DNA shotgun metagenomic pyrosequencing. In this study, *Roseburia*, *Lactobacillus*, and *Enterococcus* showed reductions, and *Coprococcus* and *Anaeroflum* were enriched in response to monensin alone or monensin in combination with virginiamycin or tylosin. Another important result was the enrichment in *E. coli* in the monensin/virginiamycin and monensin/tylosin treatments, but not in the monensin-alone treatment.

The impact of *Bacillus licheniformis* metabolites and the peptide antibiotic enramycin on the caecal microbiota was compared by Chen and Yu [2]. They reported that the diversity (richness and evenness) of bacterial species in the caeca of the chicken treated with *B. lichenofromis* metabolites was higher than in the control group. The share of obviously beneficial bacteria associated with probiotic properties, such as *Lactobacillus crispatus* and *Akkermansia muciniphila,* was also increased due to exposure of the chicken

to *B. lichenofromis* metabolites. Exposure of the broilers to enaramycin led to an elevation of *Clostridium bacterium*, *Enterococcus cecorum*, *Anaeromassilibacillus* sp., *Ruminococcus* sp. SW178, *Lachnoclostridium* sp., and *Blautia* sp. in the caecal microbiota. Noteworthy, now butyrate-producing genera *Ruminococcus* (order *Eubacteriales*, family *Oscillospiraceae*) and *Blautia* (order *Eubacteriales*, family *Lachnospiraceae*), along with *Coprococcus, Roseburia,* and *Faecalibacterium* (other representatives of the class *Closrtidia*, order *Eubacteriales*), are suggested to be favorable components of normal human column microbiota exhibiting anti-inflammatory properties [30]. A deficiency of these genera in the microbiota is associated with the progression of Parkinson's disease.

An effect of a peptide antibiotic, bacitracin, and a *Bacillus subtilis*-derived probiotic on the caecal microbiota of chickens infected with species of *Eimeria* (causative agent of coccidiosis) was described by Jia et al. [29]. The relative abundance of species *Butyricicoccus pullicaecorum*, *Sporobacter termitidis,* and *Subdoligranulum variabile* was increased in the chicken group challenged with *Eimeria*. It is known that *Butyricicoccus pullicaecorum* and *Subdoligranulum variabile* (both belong to the family *Oscillospiraceae*) produce butyrate and other short-chain fatty acids that suppress the development of *Eimeria* but are unfavorable for the microbiota [30]. *Sporobacter abundance* was shown previously to be reduced when the chickens were treated with a mixture of probiotic Bifidobacterium strains [31]. Similar effects of bacitracin and the probiotic were reported in [29].

The phytobiotics (primary or secondary components of plants that contain bioactive compounds) are proposed by Clavijo V. to be divided into four groups: (1) Herbs (products from flowering, non-woody, and non-persistent plants); (2) botanicals (whole plants or processed parts); (3) essential oils (hydro-distilled extracts of volatile plant compounds); and (4) oleoresins (extracts based on non-aqueous solvents) [1]. The beneficial impact of phytobiotics on the poultry gastrointestinal tract microbiota is reported by Hasted et al. and Abdel Baset et al. [15,32]. Abdo et al. described a cumulative effect of *B. subtilis*-derived probiotics and *Yucca shidigera* extract on water quality, histopathology, antioxidants, and innate immunity in response to acute ammonia exposure in a fish, *Oreochromis niloticus* (*Nile tilapia*). Phytobiotics are suggested to be cost-competitive but just a supplementary agent for the control of the normal and pathogenic microbiota in the chicken [33].

Significantly, the beneficial effects of the phytobiotics on the GIT microbiota and mitigating the severity of coccidiosis are attributed mostly to the immune-stimulating action of polysaccharides, flavonoids, and essential oils or to prebiotic-like action on trophic chains within the microbiota [1]. Meanwhile, the effect of the dry plant biomass may be partially explained by the presence of live bacteria in the phylloplane, which may confer a probiotic action on the GIT microbiota [34]. Members of the genus *Bacillus* and other endospore-forming aerobic bacteria (e.g., family *Paenibacillaceae,* order *Bacillales*) may be considered one of the most probable candidates to contribute to this effect for two reasons: (1) Due to the high ability of the endospores to survive drying, heating, and other unfavorable conditions upon preparation and storage; and (2) due to the commonly acknowledged beneficial impact of the probiotics derived from *Bacillus* on the normal and pathogenic microbiota of the chicken GIT. Substituting the probiotics with phytobiotics (e.g., grass meal) is able to overcome the principal shortcoming of modern commercial probiotics—the high manufacturing cost.

Taking into consideration this idea, the present study had the objective of estimating the growth-stimulating effects of the grass meal on the chicken using specimens collected in two distinct geographic locations in comparison to a negative control group (obtaining no medicines) and a positive control group fed a diet supplemented with the feeding antibiotic enrostin (100 mg/mL ciprofloxacin and $10^6$ MU/mL colistin sulfate). The bacterial load of the grass meal specimens was qualitatively and quantitatively assessed by using metagenomic sequencing of libraries obtained with Ferier_F515 and Ferier_R806 primers specific to the V4 region of 16S ribosomal DNA [35]. Particular attention was paid to endospore-forming bacteria (*Bacilli sensu lato*), which are relatively widespread in phylloplane [36] and have been suggested as efficient veterinary probiotics [2,28].

In parallel, the protective effect of the grass meal specimens against spontaneous invasion of the chickens with *Eimeria tenella* (coccidiosis) was studied. Molecular analysis by PCR with primers EtF and EtR specific to ITS-1 of the ribosome cluster [37] demonstrated the presence of this parasite in the ileum digesta of six chickens from seven dead in the negative control group in the course of the trials in the single dead chicken from experimental group KS2. One more chicken from the NC group, a chicken from the KS2 experimental group, and three chickens from the TS1 experimental group died in the course of the trials, as none of the chickens that survived until the end of the trials exhibited *E. tenella* DNA in the ileum digesta. This observation allows hypothesizing that the grass meal may confer a specific anti-coccidiosis effect on the chicken or exhibit an overall restorative effect, increasing their resistance to parasitic invasions.

## 2. Materials and Methods

### 2.1. Collection and Drying of Grass Specimens

Four specimens of the grass biomass (mostly members of the family Poaceae: *Dactylis glomerata*, *Phleum pretense,* and *Bromus inermis*) were collected in two locations in Kursk region (GPS 51.8104° N, 36.3095° E; 51.8129° N, 36.3070° E) and two locations in Tambov region (GPS 52.861625° N, 41.277611° E; 52.869416° N, 41.258822° E) during the period of May 20th to the 10th of June, 2022. A description of the locations is shown in Table 1.

**Table 1.** Description of the locations where grass specimens used for biological trials were collected.

| | Name | Region | GPS Coordinates | Description of Locality |
|---|---|---|---|---|
| 1 | KS1 | Kursk | 51.8104° N, 36.3095° E | The collection site is located in the floodplain of Vinogrobl River (Kursk, district of the Kursk region, Kamyshinsky Village Council). The height is 168 m above sea level. The slope of the terrain is insignificant or absent. The micro relief is expressed by bumps and washouts. The soil type is floodplain. Fragmentary areas of hydrophilic vegetation. The northwestern edge of the location is bounded by an alder and birch grove. |
| 2 | KS2 | Kursk | 51.8129° N, 36.3070° E | The collection site is located in the upper part of the slope formed above the -floodplain terraces of Vinogrobl River (Kursk, district of the Kursk region, Kamyshinsky Village Council). The height is 184 m above sea level. The slope of the terrain is 3–5°, falling from the northwest to the southeast. The soil is low-power chernozem. There are free-standing fruit trees (apple, pear). Farm lands are located in the immediate vicinity to the west and east of the location. |
| 3 | TS1 | Tambov | 52.861625° N, 41.277611° E | The collection site is located on the territory of agricultural land (Tambov district of the Tambov region, Lysogorsky Village Council). Soils are represented by typical chernozem. The slope of the terrain is insignificant or absent. The collection point is located at where the forest belt divides in two rows of trees, represented by poplars, between fields with agricultural crops (winter wheat). |
| 4 | TS2 | Tambov | 52.869416° N, 41.258822° E | The collection site is located on the supporting part of the ravine in front of the forest (Tambov district, Tambov region, Lysogorsky Village Council). The soil type is meadow chernozem. The forest area of the beam type. The predominant type of trees are oak; maple, alder, aspen, elm and linden. The collection site is located in a meadow with mixed grass. |

Each specimen weighed 7–10 kg. The mown grass was distributed in a thin layer on a wooden surface and dried in the open air without exposure to direct sunlight for 10–12 days. The hay was turned over daily to accelerate the drying process and avoid rot.

After drying, each hay specimen was cut with scissors into pieces below 1 cm and ground in a hand coffee grinder. Each portion was treated for 1.5–2 min while avoiding heating. The meal was kept in a plastic bag with zippers until it was used for animal trials.

### 2.2. In Vitro Testing of Grass Specimens

The total composition of the hay microbiota was determined by 16S sequencing. Briefly, 100–120 mg samples were taken from each hay specimen and put into a 1.5 mL Eppendorf tube. One hundred microliters of sterile deionized water was added, and the total DNA was isolated by using the GMO-B Sorbent Kit using CTAB as a lysing agent (Syntol, Moscow, Russia), following the manufacturer's instructions. The isolated DNA samples were sent to the State Research Institute of Agricultural Microbiology (Pushkin, Russia) for analysis. 16S DNA libraries were composed using Ferier_F515 (5′-3′) GTGCCAGCMGCCGCGGTAA and Ferier_R806 (5′-3′) GGACTACVSGGGTATCTAAT primers [37], as described below.

Total bacterial contamination and a count of *Proteobacteria* in the hay specimens were determined by microbiological methods. Fifty microliter samples of the dry grass meal were placed into a 1.5 mL Eppendorf tube with 1.000 µL of sterile deionized water, mixed intensively with vortexation, and incubated for 20 min at room temperature. Then 100-fold and 10.000-fold dilutions of the extract were prepared by the subsequent transfer of 10 µL aliquots of the initial grass extract into 1.000 µL volumes of sterile deionized water in new tubes. 10 µL aliquots of each grass extract and its 100-fold dilution were distributed onto LB agar plates (pepton bacto 10 g/L, yeast extract bacto 5 g/L, NaCl 5 g/L, agar bacto 15 g/L) for assessing total bacterial contamination and onto LB agar plates supplemented with 35 µg erythromycin for assessing *Proteobacteria* count. The plates were incubated at 30°C for 48 h, and the number of colonies was calculated manually.

The count of *Bacilli sensu lato* (number of live thermostable endospores) was determined as follows: 50 µL samples of the dry grass meal were placed into a 1.5 mL Eppendorf tube, 1.000 µL sterile deionized water was added, and the tube was incubated at 90 °C for 10 min without preliminary mixing. Then the samples were thoroughly mixed at a hand vortex and 100 times diluted by transferring 10 µL aliquots of the heated grass extract into 1.000 µL volumes of sterile deionized water in new tubes. 10 mL aliquots of each heated grass extract and its 100-fold dilution were distributed onto LB agar plates (pepton bacto 10 g/L, yeast extract bacto 5 g/L, NaCl 5 g/L, agar bacto 15 g/L). Each specimen was analyzed in duplicate. The plates were incubated at 30 °C for 48 h, and the number of colonies was calculated manually. The number of colonies in the range of 20–200 per plate was suggested to be adequate for an accurate calculation of the initial contamination of the grass sample with the endospore-forming bacteria.

Individual bacterial colonies were transferred onto LB agar plates using the triple streak exhausting method, which is used to inoculate liquid media (1 mL LB broth in 20 mL flacons with cotton plugs), which were incubated for 40 h at room temperature. The cultures were used for genomic DNA purification with GMO-B sorbent kits by Syntol (Russia), using CTAB as a lysing agent. The purified DNA in amounts of 1 µL with concentrations of 0.2–0.4 µg/µL was used as templates for PCR with primers 8F (AGAGTTTGATCCTG-GCTCAG) and 1492R (TACCTTGTTACGACTT) described earlier [34]. Dream Taq thermostable DNA polymerase (Thermo Fisher Scientific, USA) was used in the amount of 1 U per 30 µL reaction mix. The following thermal cycling parameters were applied: 94 °C–2 min; (94 °C—30 s, 60 °C—45 s, 72 °C—30 s)—30 cycles. Briefly, the 1473 bp-long PCR product was purified with the ColGen Silica Sorbent Kit by Syntol (Russia) following the manufacturer's instructions and sent for custom sequencing to Eurogen LLC (Russia, Moscow) with primers 8F, 1492R, and 926R (CCGYCAATTYMTTTRAGTTT). Three sequences covering the 16S rDNA gene were merged, and the resulting 1473–1474 bp-long

sequences were compared with NCBI GenBank using the Nucleotide BLAST utility. The name of the closest sequence and its accession numbers were fixed as the ID of the isolate.

### 2.3. Chickens and In Vivo Trials

The experimental protocol was approved at a meeting of the Local Ethics Committee of the VIGG (Protocol No. 1 dated 15 February 2018).

For the first seven days, 130 one-day-old Ross 308 cross broilers were placed in the vivarium of the Skryabin Academy of Veterinary Medicine and Biotechnology (Moscow, Russia) and kept at $32 \pm 1$ °C on a 12 h photoperiod in cage batteries with a mesh floor with an area of $80 \times 90$ cm and 20 heads per cage. The chickens had ad libitum access to water. One washer and drinker per 10 heads was used. The sex of the birds was not determined. In this period, the chickens were fed a complete starter diet "PK-5-1", purchased from Stavropolsky Kombikorm (Stavropol, Russia), without being divided into groups. During this period, they were kept in the cage.

On day 7, each chicken was weighted and distributed into one of six groups (two control and four experimental), with 20 heads in each group, using a method of pairs of analogs as described previously [24]. Ten birds not included in the experimental groups were kept in a separate cage as a reserve on a PC group diet containing toltrazuril and enrostin. They were not taken into account when the growth performance parameters were assessed and were used as a negative control for the *E. tenella* PCR diagnosis. Namely, the birds from this group were sacrificed once losses from natural causes were registered in experimental or control groups, and ileum digesta was used for DNA purification PCR and PCR with primers EtF and EtR.

All birds were kept in cages with a concrete floor with an area of 1.5 m $\times$ 2 m covered with sawdust litter, which was changed twice a week. The initial average live body weight of the chickens in the experimental and control groups at the beginning of the experiment is shown in Table 2.

**Table 2.** Experimental design of the biological trials—initial composition of the experimental groups.

| Group Name | Additive | Live Body Weight, g | | |
|---|---|---|---|---|
| | | Average | Minimal | Maximal |
| Negative control | No | 172.4 | 151 | 184 |
| Positive control | | 173.5 | 153 | 183 |
| KS1 grass meal | 1% KS1 g.m. | 173.5 | 149 | 185 |
| KS2 grass meal | 1% KS2 g.m. | 171.8 | 151 | 184 |
| TS1 grass meal | 1% TS1 g.m. | 171.7 | 152 | 185 |
| TS2 grass meal | 1% TS2 g.m. | 173.3 | 153 | 184 |

Further, the experiment was carried out until the 42nd day of life (35 days). During this period, the chickens were kept on the floor. Each group had *ad libitum* access to the food. The complete diet without antibiotics Ekorm-ROST grower diet purchased from Stavropolsy Kombikorm (Russia) was provided in excess twice a day, about 8 AM and 6 PM, each diet portion was weighted. Each experimental diet was prepared for the whole period of the experiment by adding 1% of the respective grass meal sample to the whole volume of the diet (300 g of the grass meal per 30 kg of Ekorm-ROST diet) and mixing in a 100 L hopper with a propeller stirrer from EuroPlast (Russia, Moscow). Enrostin and toltrazuril were not mixed with the diet since they were administered to PC group birds with drinking water.

Before providing a fresh diet, the residue left from the previous dosage was weighted and subtracted from the initial weight of the dosage to determine the fodder consumption and calculate the feed conversion ratio (FCR).

The chickens were weighted weekly on days 14, 21, 28, 35, and 42 of their lives (days 0, 7, 14, 21, and 28 since the start of the experiment). Each group was weighted as a whole, and the average bird mass in a group was calculated. The living weight was used as the output parameter. It was expressed as the medium arithmetical value of an average bird in each group in g and in % normalized to the positive control group. FCR was calculated as described previously [4].

Each chicken was killed within two to three hours. The dead chickens were subjected to autopsy for the purpose of collecting ileal digesta specimens, which were immediately frozen. The healthy control chickens from the reserve group were humanely killed through carbon dioxide inhalation at the same age, while spontaneous death due to contamination was registered in the negative control group. Briefly, 100 milligrams of ileal digesta was sampled in duplicate from each chick and used for DNA purification using the K-Sorb Micro-Column Sorbent Kit (Syntol, Moscow, Russia), following the manufacturer's instructions. The DNA samples isolated from the ileum digesta were analyzed by PCR with primers EtF (AATTTAGTCCATCGCAACCCT) and EtR (CGAGCGCTCTGCATACGACA) specific to ITS-1 of the ribosome cluster [38] and sequenced using the Sanger method using the BigDye Terminator v3.1 Cycle Sequencing Kit (Thermo Fisher Scientific, USA) and Nanophore 05 genetic analyzer (Syntol, Russia) once PCR products appeared. The derived sequences were compared to NCBI GenBank. Their affiliation with the genomic DNA of *Eimeria tenella* was verified by similarity with the *Eimeria tenella* genome assembly, chromosome 13 (NCBI GenBank Accession number HG994973).

At the end of the experiment, all broilers were humanely killed through carbon dioxide inhalation as described formerly [39].

### 2.4. 16S DNA Library Construction, Sequencing and Bioinformatics Analysis

A paired-end sequencing library was prepared from the PCR product obtained using the extracted DNA as a template and Ferier_F515/ Ferier_R806 primers. The Illumina Nextera XTLibrary Preparation Kit (Illumina, San Diego, CA, USA) was used for constructing the library. The library quality was assessed using a Qubit 2.0 fluorometer (Thermo Scientific, Waltham, MA, USA) and a Bioanalyzer 2100 system (Agilent, Santa Clara, CA, USA). The library was then sequenced on an Illumina NovaSeq 6000 platform (Illumina) to generate 150 bp paired-end reads. Quality control and filtering of sequenced raw reads were performed using Trimmomatic (version 0.38). A mean quality lower than Q20 in a 100 bp sliding window was considered the criterion. The reads that mapped to eukaryotic genomes on Bowtie2 (version 2.3.4.1) were filtered out. The clean reads were assembled using MEGAHIT (version 1.1.3) in pair-end mode. Bioinformatics analysis was performed using MicrobiomeAnalyst [39]. Fisher's alpha index (species richness) and Shannon index (species evenness) were used to evaluate the alpha diversity of the bacterial compositions. The overall differences in the bacterial community were analyzed through a heat map and principal coordinate analysis (PCoA) on QIIME 2 (version 2017.4). Correlation analysis was performed using Spearman's correlation coefficient and visualized using the R package "corrplot" (version 0.84).

### 2.5. Statistical Analysis

For each group, the following indicators were calculated: Average body weight (ABW), average feed intake per broiler per day (FI), average daily weight gain per broiler per day (ADG), and feed conversion index (FCR). Food intake for each group as a whole was recorded twice a day, immediately before each feeding. Dead birds were excluded from the count on the day of death.

The means and standard deviation (SD) of growth parameters (ABW, FI, ADG, and FCR) were calculated weekly and over the entire 7–42-day period as described previously, and daily means were calculated by dividing the indicator for the period by the number of days in it [24]. The Mann–Whitney U-test was used to determine the differences between

each pair of groups. It was suggested that differences be kept confidential once $p < 0.05$ was found. Statistics 8.0 for Windows was used for statistical analysis.

## 3. Results

### 3.1. Analysis Grass Metagenome Diversity

Four mixed grass samples were mowed down in two geographically remote locations in the Chernozem region of the European territory of Russia (KS1 and KS2 from Kursk region, TS1 and TS2 from Tambov region). The mown grass was dried in the open air outside in direct sunlight and milled as described in Materials and Methods. A metagenomic assay of the microbiome in the hay samples was carried out after two months of long storage of the hay meal samples at room temperature. In parallel, the same samples were subjected to biological trials on the broilers.

Metagenomic 16S rDNA-based analysis of the hay samples before filtering indicated that 53–86% of the sequences belonged to *Proteobacteria*, whereas most others were attributed to the plant mitochondrial genome. This demonstrated the insufficient specificity of the chosen primers to the bacterial genome, but it was not possible to change them for more specific variants such as 8F + 1492R or 8F + 926R due to the necessity of keeping a limited PCR product length in order to maximize read quality and coverage. Statistically treated data from metagenomics analysis of the hay specimen microbiome after subtracting the sequences belonging to plant mitochondria are shown in Figure 1.

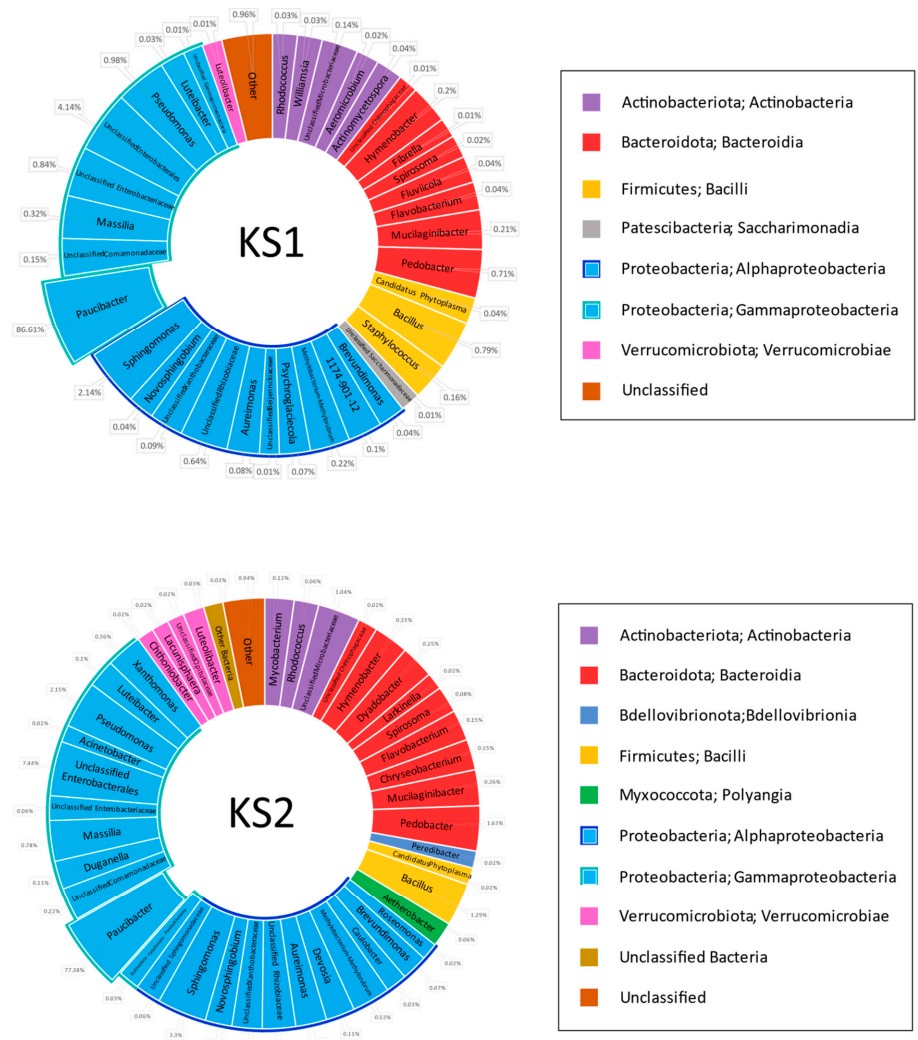

**Figure 1.** *Cont.*

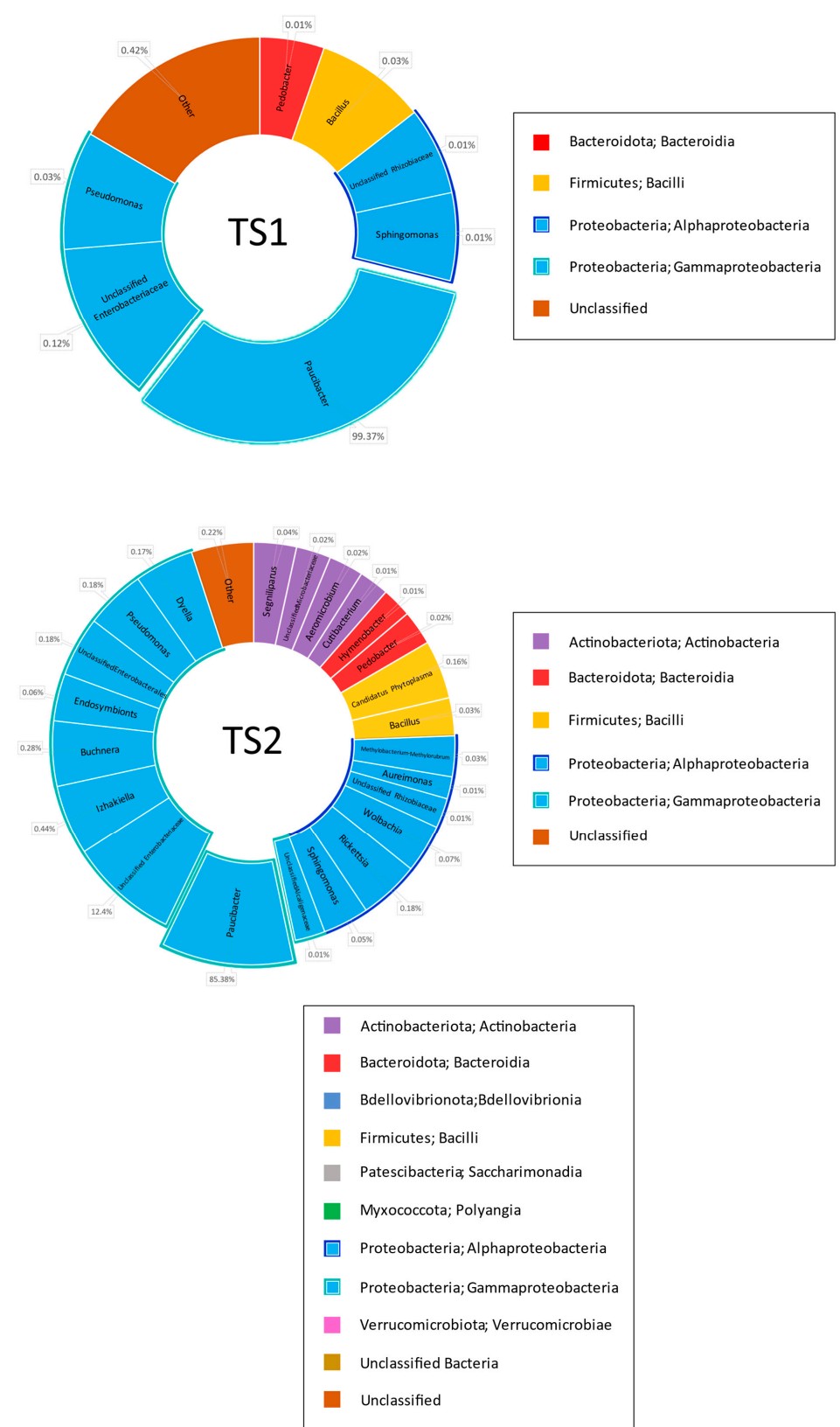

**Figure 1.** Results of metagenomics analysis of the hay samples KS1, KS2, TS1, and TS2 carried out by 16S ribosomal DNA metagenomic sequencing. Share of sequences attributed to a certain genus is shown. The affiliation of each genus to a certain class and type of bacteria is indicated using colored edging. Square of the sectors is shown proportionally to share of the taxa in the microbiome on logarithmic scale.

First of all, the analysis demonstrated an absolute dominance of the genus *Paucibacter* (*Betaproteobacteria, Burkholderiales,* and *Burkholderiales genera incertae sedis*). This genus comprises 86.6% in KS1, 77.4% in KS2, 99.4% in TS1, and 85.4% in the TS2 microbiome. Some representatives of this genus are reported as human pathogens causing bacteremia likely due to Acinetobacter and *Pseudomonas* [40], algicides [41], and agents providing degradation of cyanobacteria-derived toxins in fresh water [42]. Other genera ubiquitous within the studied cohort belonged to *Proteobacteria: Pseudomonas* (class *Gammaproteobacteria*), *Sphingomonas,* and *Aureimonas* (*Alphaproteobacteria*). The share of these genera in the microbiome reached the range of 0.03–2.15%. The total share of class *Gammaproteobacteria* comprised 93.1% in KS1, 88.6% in KS2, 99.4% in TS1, and 99.1% in TS2, and class *Alphaproteobacteria,* 3.4% in KS1, 4.7% in KS2, 0.2% in TS1, and 0.4% in TS2.

Other high-ranking bacterial taxa found in all grass specimens were *Bacteroidota* class *Bacteroidia* and *Firmicutes* class *Bacilli*. The share of *Bacteroidota* was 1.3% in KS1, 3.0% in KS2, 0.01% in TS1, and 0.03% in TS2. Class *Bacilli* comprised 1.0% in KS1, 1.3% in KS2, 0.03% in TS1, and 0.2% in TS2. The share of the genus *Bacillus* is 0.79% in KS1, 1.29% in KS2, 0.03% in TS1, and 0.03% in TS2.

Type *Actinobacteriota*, class *Actinobacteria,* was represented in all grass samples except TS1: 0.3% in KS1, 1.2% in KS2 and 0.1% in TS2.

Type Verrucomicrobiota class Verrucomicrobiae is represented in KS1 (0.01%) and KS2 (0.1%) grass samples.

Types *Myxococcota* class Polyangia and *Patescibacteria* class *Saccharimonadia* were represented in the single grass sample KS1, where they share 0.06% and 0.01% of the microbiome, respectively.

Type *Bdellovibrionota* class *Bdellovibrionia* was found in the single sample KS2, where it shared 0.02% of the microbiome.

No rare classes of bacteria are found in TS1 or TS2 samples.

Characterizing genus diversity in the grass samples, one should note that bacterial diversity in KS1 and KS2 was higher than in TS1 and TS2 samples. In turn, bacterial diversity in the TS1 sample was lower than in TS2. This observation was confirmed by the following digital values:

- Share of the dominating genus *Paucibacter*: 86.6% in KS1, 77.4% in KS2, 99.4% in TS1, and 85.4% in TS2.
- Number of classes of bacteria found is 7 in KS1, 9 in KS2, 4 in TS1 and 5 in TS2.
- Number of genera of bacteria found is 36 in KS1, 43 in KS2, 7 in TS1, and 23 in TS2.
- Subdominant taxa exceeding 1% of the microbiome are represented in the KS1 sample with unclassified Enterobacteriales (4.1%) and the genus *Sphingomonas* (2.1%); in KS2, these are unclassified Enterobacteriales (7.4%) and the genera *Sphingomonas* (3.3%) and *Pseudomonas* (2.2%).
- Subdominant taxa are absent in TS1 and TS2 samples.

A microbiological assay was used for verification of the metagenomics data. Plating non-heated extracts of the grass meal confirmed the dominance of *Proteobacteria,* which accounted for $>10^7$ c.f.u. per g in all four samples. This observation proved the essential survival of aerobic *Proteobacteria* in the dry grass biomass under storage, although they do not produce endospores. The load of spore-forming microorganisms was TS1—$6.0 \times 10^2$. TS2—$1.6 \times 10^6$. KS1—$1.0 \times 10^4$. KS2—$3.0 \times 10^4$ that did not correspond to data from metagenomics analysis (Figure 1). Apparently, the high discrepancy between data obtained by metagenomics and microbiological methods may be explained by a high share of vegetative (perhaps not alive) cells and a low share of thermostable endospores of *Bacilli sensu lato* in the TS2 sample, whereas the KS1 and KS2 samples contained *Bacilli* mostly in the form of thermostable endospores. The species specificity of the isolated bacterial clones was determined by molecular methods (16S rDNA sequencing). The results of this analysis are shown in Table 3.

**Table 3.** Results of molecular identification of species specificity of bacterial clones isolated from dry grass samples.

| Isolate Number | Selective Factor | Closest Relative Up on 16S rDNA Sequencing; NCBI GenBank acc. # | Coincidence with the Closest Relative, % |
|---|---|---|---|
| | | Host grass sample KS1 | |
| Ke1-1 | Erythromycin | *Pantoea agglomerans*; MT634720.1 | 99 |
| Ke1-2 | Erythromycin | *Acinetobacter lwoffii*; MT626722.1 | 99 |
| Ke1-3 | Erythromycin | *Acinetobacter lwoffii*; P054803.1 | 99 |
| Kp1-1 | No | *Bacillus subtilis*; KY780544.1 | 99 |
| Kp1-2 | No | *Bacillus velezensis*; OP550066.1 *Bacillus amyloliquefaciens*; MT889680.1 | 99 |
| Kc1-1 | Heating at 90 °C | *Bacillus subtilis*; JX164089.1 | 99 |
| Kc1-2 | Heating at 90 °C | *Bacillus velezensis*; OP550068.1 | 100 |
| Kc1-3 | Heating at 90 °C | *Bacillus velezensis*; OP550066.1 | 99 |
| Kc1-4 | Heating at 90 °C | *Bacillus altitudinis*; MN640111.1 | 95 |
| Kc1-5 | Heating at 90 °C | *Bacillus subtilis*; MH588270.1 | 99 |
| Kc1-6 | Heating at 90 °C | *Paenibacillus dendritiformis*; OP811875.1 | 84 |
| Kc1-7 | Heating at 90 °C | *Bacillus tequilensis*; KM374724.1 | 97 |
| Kc1-9 | Heating at 90 °C | *Bacillus siamensis*; OP714409.1 | 97 |
| | | Host grass sample KS2 | |
| Ke2-1 | Erythromycin | *Pseudomonas koreensis*; OP550075.1 | 95 |
| Ke2-2 | Erythromycin | *Pantoea agglomerans* MT605812.1 | 100 |
| Kc2-2 | Heating at 90 °C | *Bacillus velezensis*; MN559570.1 | 98 |
| Kc2-3 | Heating at 90 °C | *Bacillus subtilis*; KX161426.1 | 98 |
| Kc2-4 | Heating at 90 °C | *Bacillus licheniformis*; OM267735.1 | 99 |
| | | Host grass sample TS1 | |
| Te1-1 | Erythromycin | *Pseudomonas fulva*; KP761419.1 | 95 |
| Tc1-1 | Heating at 90 °C | *Bacillus inaquosorum*; KT720186.1 | 99 |
| Tc1-2 | Heating at 90 °C | *Bacillus subtilis*; MN631031.1 | 97 |
| | | Host grass sample TS2 | |
| Te2-3 | Erythromycin | *Bacillus stercoris*; MN704461.1 | 95 |
| Tc2-1 | Heating at 90 °C | *Bacillus subtilis*; MW898140.1 | 97 |
| Tc2-2 | Heating at 90 °C | *Bacillus subtilis*; HM480329.1 | 96 |
| Tc2-3 | Heating at 90 °C | *Bacillus safensis*; MN215310.1 | 93 |
| Tc2-4 | Heating at 90 °C | *Bacillus mojavensis*; KY622215.1 | 98 |
| Tc2-5 | Heating at 90 °C | *Bacillus subtilis*; KU667124.1 | 95 |

Data from Table 3 confirm the conclusion about the dominance of *Gammaproteobacteria* in the microbiome of all grass samples; however, the exact species specificity of the isolates differs from that determined by metagenomics assay in the following parameters:

No representatives of the genera *Paucibacter* and *Sphingomonas* were isolated from any grass samples. Apparently, they are unable to grow fast at LB media under the chosen conditions. In contrast, isolated representatives of *Gammaproteobacteria* belonged to the genera *Pantoea*, *Acinetobacter*, and *Pseudomonas*, which shared just a small amount of the

consortium detected by metagenomics assay. *Pseudomonas* is the only genus found in the grass sample after metagenomics analysis and microbiological seeding.

The species affiliation of *Bacilli* sensu lato was overall similar (*B. subtilis* and close species *B. velezensis, B. amyloliquefaciens, B. altitudinis, B. tequilensis, B. siamensis, B. licheniformis, B. inaquosorum, B. stercoris, B. safensis,* and *B. mojavensis*) in all samples, although their quantitative representation fluctuated in the range of 4 orders. The only isolate containing thermostable endospores not belonging to the group of mesophilic *bacilli* was isolated from a KS1 grass sample. It has 16S rDNA that is 84% similar to *Paenibacillus dendritiformis* (family *Paenibacillaceae*, not *Bacillaceae*).

For isolating *Bacilli sensu lato*, 50 mg of ground grass samples was placed in 1.7 mL Eppendorf tubes, flooded with 1 mL deionized water, and heated for 10 min in a solid-state thermostat for the tubes. The tubes were not mixed before heating to avoid casting the bacteria-containing material onto the not-heated lid. After the heating, the samples were thoroughly mixed by vortexation, and 50 µL aliquots of the suspension were picked up by a trimmed 200 µL automated pipet tip, placed onto the nutrient agar, and distributed by a glass spatula. For isolating *Proteobacteria*, the ground grass suspensions prepared in a similar way without heating were picked up by a trimmed 200 µL automated pipette tip, placed onto the nutrient agar containing 35 µg/µL erythromycin, and distributed by a glass spatula. The respective experimental procedures are described in detail in Section 2.1.

Interestingly, a *Bacillus stercoris* colony was found in the TS2 sample under erythromycin selection without heating. Mesophilic *bacilli* were found in the TS1 sample, although metagenomic assay demonstrated the complete absence of this group in the microbiome. This observation makes doubtful the accuracy of the metagenomics data, apparently due to incomplete DNA extraction from the grass sample.

No *B. cereus* was found among *Bacilli sensu lato* isolates from the TS2 sample, although following metagenomics data, this species significantly outnumbered *B. subtilis, B. stercoris, B. safensis,* and *B. mojavensis* in this source. This fact admits the assumption that DNA is poorly isolated from thermostable spores, whereas a major share of *Bacilli sensu lato* remains on the grass in the vegetative form, which does not survive heating at 90 °C.

Isolates Kp1-1 and Kp1-2 from the KS1 sample under non-selective conditions belonged to mesophilic *bacilli*, not *Proteobacteria,* although following metagenomic data, the share of this group in the KS2 metagenome is ~1%. This is the clearest evidence of a substantial bias that appeared at the stage of DNA purification and 16S rDNA metagenomic analysis.

In our opinion, the data from the microbiological assay of thermostable endospores were the most accurate. They were first used for the characterization of the tested grass samples.

### 3.2. Biological Trials of the Additives on the Basis of the Grass Meal

Chickens of a rapidly growing cross, Ross 308, were used in the experiment. As shown in Table 2, the broad-spectrum antibiotic enrostin (complex preparation containing 100 mg ciprofloxacin (fluoroquinolone) and 10 mU colistin (peptide ionophore)) was used for preventing bacterial infection in all flocks in periods 1–7 of the life prior to forming experimental groups. No bird losses were registered at this time. On day 7 of life, the flock was distributed into the groups using a method of pairs of analogs for equilibrating the average mass of the birds in each group, as shown in Table 2. Then the positive control group (PC) was treated with Stop-Coccid (days 14–16–25 µg/mL toltrazuril within the drinking water; days 28–33—enrostin within the drinking water). The negative control group (NC) in periods 8–42 obtained no additives, and the experimental groups permanently obtained 1% grass meal within the food.

In the period of days 14–21, a total of 7 chickens were lost under these conditions in the NC group (35% mortality). Two chickens died at the same time in the KS2 experimental group (10% mortality), and three chickens died in the TS1 experimental group (15% mortality). No losses were registered in the KS1 and TS2 groups or in the positive control group (PC). This observation gives evidence that the grass meal may partially or

completely substitute the antibiotic enrostin and the coccidiostatic preparation toltrazuril as a means of protecting the chickens from death caused by infection or invasion.

Before the death, all seeking chickens exhibited the following symptoms: Ruffled feathers, diarrhea, and difficult gait. Their feces had a peculiar acidic smell. One chicken had liquid discharge from its beak. The visual examination of the inner organs of dead chickens after autopsy elucidated an edema and inflammation of the small intestine, and particularly the caeca in all birds. DNA was isolated from the ileum digesta and analyzed by PCR with primers EtF (AATTTAGTCCATCGCAACCCT) and EtR (CGAGCGCTCTGCATACGACA) specific to ITS-1 of the ribosome cluster [37]. The PCR was positive in the samples from six of the seven chickens that died in the NC group and in the single dead chicken from the experimental group KS2 (Table 4). The PCR products were sequenced by the Sanger method. The derived sequences were compared to NCBI GenBank and exhibited 100% similarity with the *Eimeria tenella* genome assembly, chromosome 13 (NCBI GenBank Accession number HG994973). Taken together, these data unambiguously prove that the death from coccidiosis (spontaneous invasion with *E. tenella*) in the NC group attained 37%, and these losses were completely prevented by either the combination of toltrazuril and enrostin (positive control group) or by grass meal (experimental groups KS1, TS1, and TS2) and partially prevented in the KS2 group. No PCR products with primers EtF and EtR were found in the ileum digesta DNA of the reserve group (totaling 10 heads), which were sacrificed simultaneously with the birds that died from the invasion.

**Table 4.** The chicken losses in the course of the experiment and *E. tenella* diagnosis in their ileal digesta samples.

| Group | Number of the Lost Chickens in Period 14–21 Days of Life | Number of Positive Responses during *E. tenella* DNA Screening of the Ileum Digesta Samples |
|---|---|---|
| NC (20 chickens) | 7 | 6 |
| KS2 (20 chickens) | 2 | 1 |
| TS1 (20 chickens) | 3 | 0 |
| Reserve group (10 heards) | - | 0 |

Besides protection from invasion and infection, the impact of the grass meal additives on the average weight gain on days 14, 21, 28, 35, and 42 was determined, and the FCR coefficient was calculated. A clear lag in the NC group in comparison to the PC group was found on days 21, 28, 35, and 42 ($p < 0.01$). At the end of the experiment, the average body weight (ABW) in the NC group (2473 g) was 13% less than in the PC group (2841 g). Differences in ABW between the PC group and the experimental groups were not confidential (less than ±1% at any time during the experiment). These data prove that the tested grass meal additives are able to substitute the chemical preparations toltrazuril and enrostin as growth promoters, not as only anti-parasitic means.

Dynamics of ABW, feed intake, ADG, and FCR values in the experimental groups are shown in Table 5.

**Table 5.** Live body weight values (average weight per head, g)/FCR values of the chickens in the experimental groups.

| Groups | Parameter | Days of the Chickens Life | | | | | | |
|---|---|---|---|---|---|---|---|---|
| | | 7 | 14 | 21 | 28 | 35 | 42 | Mean ± SD |
| Negative control | ABW, g Mean ± SD | 172.4 ± 8.3 | 457.6 ± 5.3 | 928.8 ± 16.0 [a] | 1334.4 ± 11.4 [a] | 1910.4 ± 41.5 [a] | 2472.7 ± 23.5 [a] | 1212.7 ± 874.3 |
| | Feed intake, g | 33.08 | 130.41 | 279.49 | 364.09 | 510.34 | 646.92 | 327.4 ± 230.1 |
| | ADG | - | 40.74 | 67.31 | 57.51 | 82.29 | 80.33 | 65.6 ± 17.2 |
| | FCR | 1.47 | 1.89 | 1.85 | 1.92 | 2.02 | 2.17 | 1.89 ± 0.23 |

**Table 5.** *Cont.*

| Groups | Parameter | Days of the Chickens Life | | | | | | |
|---|---|---|---|---|---|---|---|---|
| | | 7 | 14 | 21 | 28 | 35 | 42 | Mean ± SD |
| Positive control | ABW, g Mean ± SD | 173.5 ± 9.1 | 462.9 ± 3.0 | 954.3 ± 8.8 [b] | 1560.7 ± 14.8 [b] | 2041.0 ± 47.1 [b] | 2840.7 ± 33.4 [b] | 1338.9 ± 1007.1 |
| | Feed intake, g | 34.59 | 124.62 | 252.35 | 428.08 | 588.98 | 748.75 | 362.9 ± 276.1 |
| | ADG | - | 41.34 | 70.20 | 86.63 | 68.61 | 114.24 | 76.2 ± 26.8 |
| | FCR | 1.4 | 2.11 | 2.1 | 1.91 | 1.87 | 1.83 | 1.87 ± 0.26 |
| KS1 grass meal | ABW, g Mean ± SD | 173.5 ± 9.0 | 462.4 ± 4.2 | 956.0 ± 12.0 [b] | 1527.6 ± 17.1 [b] | 2034.6 ± 26.3 [b] | 2851.4 ± 21.8 [b] | 1334.3 ± 1008.0 [b] |
| | Feed intake, g | 34.22 | 130.48 | 247.19 | 397.19 | 534.81 | 761.31 | 350.9 ± 269.8 |
| | ADG | - | 41.27 | 70.03 | 81.66 | 72.43 | 116.69 | 76.4 ± 27.6 |
| | FCR | 1.46 | 1.98 | 1.81 | 1.82 | 1.84 | 1.87 | 1.80 ± 0.18 |
| KS2 grass meal | ABW, g Mean ± SD | 171.8 ± 10.3 | 461.9 ± 3.1 | 952.6 ± 9.9 [b] | 1533.8 ± 11.2 [b] | 2043.0 ± 47.9 [b] | 2849.1 ± 25.6 [b] | 1335.4 ± 1009.4 |
| | Feed intake, g | 33.65 | 127.74 | 252.98 | 405.36 | 537.02 | 756.99 | 352.3 ± 260.9 |
| | ADG | - | 41.44 | 70.10 | 83.03 | 72.74 | 115.16 | 76.5 ± 26.6 |
| | FCR | 1.43 | 1.94 | 1.86 | 1.85 | 1.84 | 1.86 | 1.80 ± 0.18 |
| TS1 grass meal | ABW, g Mean ± SD | 171.7 ± 10.1 | 463.2 ± 2.8 | 953.7 ± 10.3 [b] | 1537.4 ± 12.1 [b] | 2053.6 ± 42.9 [b] | 2833.8 ± 17.4 [b] | 1335.6 ± 1006.2 [b] |
| | Feed intake, g | 34.21 | 127.91 | 246.54 | 404.11 | 536.88 | 727.91 | 346.3 ± 260.9 |
| | ADG | - | 41.64 | 70.07 | 83.39 | 73.74 | 111.46 | 76.6 ± 25.2 |
| | FCR | 1.46 | 1.93 | 1.81 | 1.84 | 1.83 | 1.8 | 1.78 ± 0.16 |
| TS2 grass meal | ABW, g Mean ± SD | 173.3 ± 8.3 | 460.9 ± 4.8 | 955.4 ± 9.2 [b] | 1533.2 ± 20.4 [b] | 2033.7 ± 23.5 [b] | 2831.4 ± 15.4 [b] | 1331.3 ± 1002.4 [b] |
| | Feed intake, g | 34.31 | 135.16 | 262.79 | 429.29 | 578.16 | 764.37 | 367.3 ± 276.3 |
| | ADG | - | 41.09 | 70.64 | 82.54 | 71.50 | 113.96 | 76.0 ± 26.2 |
| | FCR | 1.46 | 2.05 | 1.93 | 1.96 | 1.99 | 2.03 | 1.90 ± 0.22 |

[a]—statistically confident difference in the parameter value in comparison to the positive control group at the same time point of the experiment ($p < 0.05$ according to the Mann–Whitney test). [b]—statistically confident difference in the parameter value in comparison to the negative control group to the same time point of the experiment ($p < 0.05$ according to the Mann–Whitney test).

## 4. Discussion

The biological trials carried out demonstrated a clear beneficial effect of the KS1 and TS2 grass meal additives in the ABW of the chickens. In contrast, they did not exhibit an impact on daily feed intake or FCR parameters. Noteworthy, this additive completely prevented the chicken's death. In this respect, they were not inferior in effectiveness to the combination of toltrazuril and the antibiotic enrostin. On the contrary, the feed additives based on grass flour (KS2 and TS1) did not provide complete protection for chickens from *E. tenella* invasion, although mortality in these groups was lower than in the negative control group (3 and 2 dead chickens, respectively, versus 7 in the NC group). Taken together, this observation allows hypothesizing that herbal flour itself has a protective effect that suppresses the invasion of *E. tenella* in chickens, but the content of bacteria within significantly affects physiological properties when the grass sample is used as a feed additive.

Studies of the effect of herbal flour-based additives on ABW and ADG indicators have confirmed their ability to replace an antibiotic and an antiparasitic drug when feeding chickens. Starting from the 28th day of the chicks' life and up to the end of the experiment (42 days of life), the ADG index in the experimental groups KS1, TS1, KS2, and TS2 significantly differed from the NC group but not from the PC group.

At the same time, the FCR indicators in the PC group on the 21st day of the chicks' lives differed from those in the NC group for the worse. This shows that the use of a combination of toltrazuril and enrostin during this period had a negative effect on feed conversion. At the same time, the herbal meal did not have a depressing effect on the digestibility of the feed, successfully coping with the function of preventing bird death from *E. tenella* invasion (35% dead birds in the NC group vs. 0–15% in the PC and the

experimental groups). This observation shows the advantage of herbal flour enriched with certain types of bacteria as a means of protecting chickens from coccidiosis. At the same time, on the 35 and 42 days of chick life, the NC group already showed a statistically confidential lag in ADG from the PC and experimental groups KS1, KS2, and TS1. On the contrary, the KS2 group on the 35 and 42 days of the experiment showed the worst ADG index compared to the PC group.

Explaining the results obtained, we draw attention to the fact that in the TS2 herbal flour sample, with a high total content of *bacilli* potentially capable of acting as a probiotic (as derived from results of bacteriological seeding after heating), the absolute dominance of the *B. cereus* species (following metagenomics analysis), described as a conditional pathogen of chickens and other animals, was observed [43]. This species may cause a depressing effect on the assimilation of feed by chickens, along with an antagonistic effect on *E. tenella* and bacterial pathogens in the gastrointestinal tract of chickens. On the other hand, no isolates of *B. cereus* were found among the bacterial clones subjected to individual molecular typing (Table 1).

Noticeable differences in the protective effectiveness of KS1 and KS2 samples are difficult to explain since the composition of the microbiome of these samples is highly similar. The share of *bacilli* is 0.79% for KS1 and 1.29% for KS2 when assessed by metagenomic analysis. According to the microbiological seeding data, the endospore content in these samples was $1.0 \times 10^4$ in KS1 and $3.0 \times 10^4$ in KS2. Both methods give similar figures, and this does not allow considering the difference in the content of *bacilli* or their endospores as a key parameter affecting the protective effectiveness against coccidiosis in chickens. It is possible that the low protective effectiveness of the KS1 sample is due to the content of potential pathogens of chickens in it, representatives of the *Saccharimonadia* class (0.01% of the microbiome) or the genus Staphylococcus (0.16% of the microbiome); both of these groups were completely absent in the KS2 sample. There is evidence in the literature about the possibility of the presence of representatives of these groups in the human intestine and the association of their increased proportion in the microbiome with an unfavorable prognosis for diseases [44,45]. The TS2 sample exhibiting a high protective activity, along with the KS1 sample, contained few representatives of *Bacilli sensu lato* (0.3%, as shown by metagenomic analysis). We hypothesize that its beneficial impact on chicken safety during the spontaneous *E. tenella* invasion may be explained by the high share of the unclassified Enterobacteriaceae group in the TS2 sample.

Analysis of previously published data gives a number of indirect clues confirming the efficiency of grass biomass as a source of beneficial bacteria (analogs of probiotics). Probiotics, like antibiotics (e.g., enramycin and tylosin), confer resistance against *Eimeria* on the chicken, although they do not exhibit antagonism towards *Apikomplexa sporozoits* in vitro [27]. Moreover, the concept of the necessity of long-term persistence of the probiotics in GIT was revised. Obviously, the high efficiency of *Bacillus* strain's anti-pathogenic and growth-promoting effects on the chicken was acknowledged, although a pure aerobic metabolism does not allow *Bacilli* vegetate under the chicken's GIT anaerobic conditions. Moreover, extracellular culture medium from *Bacillus licheniformis* and *Bacillus subtilis* conferred a favorable impact on the GIT microbiota and daily weight gain of the broilers [2,28]. This confirms that the short-term influence of the probiotic-derived metabolite is sufficient for the favorable action of the overall probiotic. Therefore, the kinetics of the anti-pathogenic action of the antibiotics and probiotics may be more similar than previously suggested.

The effects of antibiotics and probiotics on the GIT microbiota in chickens were extensively studied by using metagenome sequencing (amplified libraries of 16S rDNA gene fragments were sequenced on the Illumina platform) [29,45]. It has been indicated that in the caeca of broilers, Clostridia are the predominant organisms [30], while the genus *Lactobacillus* is dominant in the ileum [46].

Importantly, an impact of the antibiotics monensin, virginiamycin, and tylosin on the microbiome of caeca was described in [31]: The effect of the coccidiostat monensin and the growth promoters virginiamycin and tylosin on the caeca microbiome and metagenome

of broiler chickens, 16S rRNA, and total DNA shotgun metagenomic pyrosequencing. In this study, *Roseburia*, *Lactobacillus*, and *Enterococcus* showed reductions, and *Coprococcus* and *Anaeroflum* were enriched in response to monensin alone or monensin in combination with virginiamycin or tylosin. Another important result was the enrichment in *E. coli* in the monensin/virginiamycin and monensin/tylosin treatments, but not in the monensin-alone treatment.

The impact of *Bacillus licheniformis* metabolites and the peptide antibiotic enramycin on the caecal microbiota was compared by Chen and Yu [2]. They reported that the diversity (richness and evenness) of bacterial species in the caeca of the chicken treated with *B. lichenofromis* metabolites was higher than in the control group. The share of obviously beneficial bacteria associated with probiotic properties, such as *Lactobacillus crispatus* and *Akkermansia muciniphila,* was also increased due to exposure of the chicken to *B. lichenofromis* metabolites. Exposure of the broilers to enramycin led to an elevation of *Clostridium bacterium*, *Enterococcus cecorum*, *Anaeromassilibacillus* sp., *Ruminococcus* sp. SW178, *Lachnoclostridium* sp., and *Blautia* sp. in the caecal microbiota. Noteworthy, now butyrate-producing genera *Ruminococcus* (order *Eubacteriales*, family *Oscillospiraceae*) and *Blautia* (order *Eubacteriales*, family *Lachnospiraceae*), along with *Coprococcus*, *Roseburia*, and *Faecalibacterium* (other representatives of the class *Closrtidia*, order *Eubacteriales*), are suggested to be favorable components of normal human column microbiota exhibiting anti-inflammatory properties [29]. A deficiency of these genera in the microbiota is associated with the progression of Parkinson's disease.

An effect of a peptide antibiotic, bacitracin, and a *Bacillus subtilis*-derived probiotic on the caecal microbiota of chickens infected with species of *Eimeria* (causative agent of coccidiosis) was described by Jia et al. [29]. The relative abundance of species *Butyricicoccus pullicaecorum*, *Sporobacter termitidis,* and *Subdoligranulum variabile* increased in the chicken group challenged with *Eimeria*. It is known that *Butyricicoccus pullicaecorum* and *Subdoligranulum variabile* (both belong to the family *Oscillospiraceae*) produce butyrate and other short-chain fatty acids that suppress the development of *Eimeria* but are unfavorable for the microbiota [47]. *Sporobacter abundance* was shown previously to be reduced when the chickens were treated with a mixture of probiotic Bifidobacterium strains [48]. Similar effects of bacitracin and the probiotic were reported in [29].

Chicken gut microbiota (feces) responses to *B. subtilis* probiotics in the presence and absence of *E. tenella* infection are reported by Memon F.U. [49]. The feces of the healthy control group contained about 95% *Firmicutes*, 4% *Proteobacteria,* and 1% other phyla. Infection with *Eimeria* decreased the share of *Firmicutes* to 70%, whereas *Proteobacteria* shared 21% and *Bacteroidetes* 8% of the fecal microbiome. Treatment of the healthy chicken flock with the probiotic somewhat increased the share of *Proteobacteria*, *Bacteroidetes,* and other phyla in comparison to the non-treated group. Administration of the probiotic to the chicken challenged with *E. tenella* did not affect the ratio of different bacterial phyla in the fecal microbiota, although it substantially mitigated the morbidity of the disease. The relative abundances of *Lactobacillus* within the *Firmicutes* clade accounted for 36.56%, 56.42%, 49.73%, and 54.76 in the respective groups of chickens. Escherichia-Shigella accounted for 4.42%, 25.82%, 6.41%, and 28.20% within the *Proteobacteria* clade. In contrast, decreased abundances of Kurthia, *Ruminococcus torques*, and *Clostridium* were found in *Eimeria*-infected groups compared to the healthy control group. Probiotic-treated and challenged chickens, on the other hand, restored (increased) the abundances of *Clostridium sensu stricto*, *Corynebacterium*, *Enterococcus*, *Romboutsia,* and *Subdoligranulum* and decreased the abundances of *Faecalibacterium*, *Lachnoclostridium*, *Eisenbergiella*, *Sellimonas*, *Flavonifractor*, *Monoglobus*, *Lachnospiraceae*, *Blautia*, *Ruminococcus torques*, *Christensenellaceae*, *Eubacterium hallii,* and Paludicola compared to the *Eimeria*-infected non-treated group.

Khogali reported changes in the microbiota of feces in old laying hens induced by the administration of *Clostridium butyricum* and *B. subtilis*-derived probiotics [24]. Noteworthy, the exposure of the hen to the probiotics reduced the share of pimpled eggs, a substantial share of which compromises the economic efficiency of the elderly hens. In contrast to the

caecal microbiota, the healthy hen feces contain above 85% *Firmicutes* (>98% *Lactobacillales*), 6% *Proteobacteria*, and 2% *Actinobacteria*. In old hens prone to laying pimpled eggs, above 70% of the feces microbiota is occupied by *Proteobacteria,* and the share of *Bacteroidetes* attains 4–5%, whereas the share of *Firmicures* is decreased to 15% (share of *Lactobacillales* is ~50%). Application of the bifunctional probiotic increases the share of *Firmicutes* to ~70%, reduces the share of *Proteobacteria* and *Bacteroidetes* to the normal level, and increases the share of *Actinobacteria* to 7%. It increases the share of Verrucomicrobia to 2.5%, while the contents of this group in the feces of non-treated hens are negligible. However, the share of *Lactobacillales* within *Firmicutes* after exposure to the probiotic was far from normal (15–20%). Taken together, one should conclude that probiotics are now considered a powerful tool comparable to antibiotics in terms of impact on the normal and pathogenic components of the chicken GIT microbiota and safety, but are less affordable for practical use due to a high manufacturing cost [1].

## 5. Conclusions

Concluding the analysis of the obtained results, it should be noted that they convincingly demonstrate the beneficial impact of the dry plant biomass (a mix of *D. glomerata*, *P. pretense,* and *B. inermis*) as the growth promoter when added to the food in a ratio of 1% of the diet weight. This effect was not worse than the effect of enrostin, which is traditionally used at Russian industrial poultry plants in this role. Enrostin added to the chicken food together with tolatrzuril elevated ADG up to 14.9% in comparison to the same diet without medicines. The tested dry grass biomass samples collected in different locations increased ADG to 14.6–15.2% in comparison to the negative control. Dry grass biomass is obviously more economical and safe for chickens and chicken meat consumers in comparison to any antibiotic, including enrostin.

Moreover, due to an extensive outbreak of coccidiosis that occurred in 2022, we faced a spontaneous invasion of *E. tenella* in the experimental and negative control groups and registered the efficiency of two dry grass biomass samples from four tested against the parasite invasion. We hypothesize that this effect was caused by the different microbial composition of the grass biomass. The most protective samples, KS1 and TS2, contained 0.79% *Bacilli* sensu lato, whereas the KS2 sample contained 12.3% unidentified Enterobacteriaceae. The KS2 sample, which contained the highest share of *Bacilli* sensu lato and was considered the most probable analog of probiotics, exhibited poor protection against mortality. We suppose that differences between KS1 and KS2 samples can be explained by differences in the prevalence of *Bacillus* species, namely, a high share of an opportunistic animal pathogen, *B. cereus,* in the KS2 sample, whereas *B. subtilis* group species *Bacillus velezensis, Bacillus amyloliquefaciens, Bacillus subtilis, Bacillus altitudinis, Bacillus subtilis, Bacillus tequilensis,* as well as *Paenibacillus dendritiformis,* dominated in the KS1 sample. We suggest that these bacteria, along with herbal bio-constituents, contribute to the suppression of opportunistic pathogens in the chicken ileum and other GIT sections. We hypothesize that these bacteria can suppress *Eimeria* egg germination, mitigating the risk of parasite invasion and the death of the bird from it, although this hypothesis still requires experimental verification.

**Author Contributions:** E.S.B. collected grass samples in Tambov region and carried out their microbiological study; M.A.D. carried out biological trials on the broiler chicken model; M.S.S. carried out metagenomic and Sanger sequencing; E.V.T. prepared illustrations and carried out statistical analysis of the data; H.T.N. purified DNA samples for molecular analysis; S.N.P. prepared grass samples in Kursk region; S.V.A. provided the chicken flock care; A.B.S. proposed the concept of the study drafted Background section and provided overall design of the manuscript. All authors have read and agreed to the published version of the manuscript.

**Funding:** The work was funded by grant of Russian Ministry of Science and Highest Education, agreement # 075-15-2021-1395.

**Institutional Review Board Statement:** The experimental protocol was approved at a meeting of the Local Ethics Committee of the VIGG (Protocol No. 1 dated 15 February 2018).

**Informed Consent Statement:** Not applicable.

**Data Availability Statement:** No supplementary data are available.

**Acknowledgments:** The authors bring thanks to Ruslan O. Aliev for assistance in preparing illustrations.

**Conflicts of Interest:** The authors declare no conflict of interest.

## Abbreviations

Added body weight (ABW), feed conversion ratio (FCR), gastric intestinal tract (GIT), grass specimen 1 from Kursk (KS1), grass specimen 2 from Kursk (KS2), grass specimen 1 from Tambov (TS1), grass specimen 2 from Tambov (TS2), internal transcribed spacer (ITS), negative control group (NC), positive control group (PC), principal coordinate analysis (PCoA), ribosomal DNA (rDNA), standard error of the mean (SEM).

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
