# Peer review of "Grass Meal Acts as a Probiotic in Chicken"

_2036-7481, doi:10.3390/microbiolres14040113_

Round 1

Reviewer 1 Report

Work is very interesting. I have not editorial and methodological comments.

Author Response

Dear Reviewer,

We a greatly appreciated with your high estimate of our MS.

Thank you so much!

On behalf of the author team,

Sincerely Elena

Reviewer 2 Report

Abstract:

Major comments:

In the abstract grouping is not clear. Authors should indicate how many groups enrolled in this study, and the numbers and ages of animals. In addition, authors should indicate the duration of the study and introduce sampling parameters in the abstract. The authors mentioned that data about the environmental factors and seasons remain insufficient but they did not provide any data in the conclusion about this question; Therefore, it is recommended that changing the paragraph with other data or at the conclusion it is essential to provide some answer to the question.

Minor comments:

Line 15: Authors should provide species of grass.

Line 16: Authors should provide full words for NGS

Line 17: Authors mentioned: 1% of the grass meal. Do the authors mean 1% of dry matter in the diet?

Line 18: Please indicate the route of administration of medicines (toltrazuril and Enrostin)

Line 19: Please indicate the dosage of Enrostin (antibiotic)

Line 23: Authors mentioned that: grass meal may substitute the antibiotic for protection against parasitic invasion. But antibiotics are commonly used against bacteria, not parasite infection.

Introduction:

Major comments:

The introduction is very long with a huge explanation and data; therefore, authors should minimize the introduction to 1000-1500 words. There are many linguistic and punctuation errors in the introduction.

Minor comments:

Line 29: Authors mentioned that: Poultry farming is the most efficient meat-producing system. It is not clear that they compare the meat-producing system in poultry farming to what system?

Line 34: please change “42nd day of life” to 42 days of life

Line 35 and 36: please rephrase

Line 39-41: Please provide a reference for the statement “Antibiotics are commonly used at low doses for infectious disease prevention in broilers, thereby ameliorating their growth and preventing losses”.

Line 47: Please provide the reference for the statement “global consumption of antimicrobials was expected to be increased by 67%, from 63,151 ± 1,560 tons to 105,596 ± 3,605 tons between 2010 and 2023.

Line 54: As a suggestion you can use the following reference:

Mirzaei, A., Razavi , S. A., Babazadeh, D., Laven, R., & Saeed, M. (2022). Roles of Probiotics in Farm Animals: A Review. Farm Animal Health and Nutrition, 1(1), 17–25. https://doi.org/10.58803/fahn.v1i1.8

Line 87: in the statement “impact of Following preparations “the following should start with a lowercase letter.

Lines 87-90: The paragraph should be rephrased.

Methods and materials:

Major comments:

In the materials and methods authors did not indicate how many replications they had in this trial. The authors should provide the chemical composition of the starter diet and grass meal. Authors should indicate the parameters that they measured in this study in the Chickens and biological trials section. The Mann–Whitney U test is a nonparametric test to compare outcomes between two independent groups. This study contains more than two groups. What test authors used to evaluate the significance between the groups of parameters such as weight gain in the study?

In the materials and methods authors should provide data that how animals were infected to Eimeria.

Minor comments:

Line 210-211: Please provide a reference for primers

Line 249: Please indicate the database and move the web address to the reference section.

Line 257: Please indicate the country of the academy

Line 258: Please indicate the size of the cage.

Line 258: The authors indicate that “The chickens had ad libitum access to water”. what about feed?

Line 264: Please provide a reference for the method of allocating the animals (pairs of analogues)

Line 271: Please indicate the exact time of providing a diet to chickens.

Lines 281-285: Please provide the statistical analysis in a separate subsection.

Line 282: The Mann–Whitney U test is a nonparametric test to compare outcomes between two independent groups. This study contains more than two groups. What test authors used to evaluate the significance between the groups of parameters such as weight gain in the study?

Line 288: Please provide a reference for the method of sacrificing the animals.

Results:

Major comments:

In the results, part authors provide illustrative figures and tables that are very valuable.

Tables should be self-explanatory. In Table 3 please define the abbreviations in the table footnotes.

In table 3, in the fourth column % should be in brackets.

Minor comments:

Lines 318-324: This paragraph is the repetition of the material and methods (not necessary to mention).

Line 443: Please remove one of the brackets [))]

In Table 4, please define all the groups under the table.

In Table 5: please provide full words in the table title and define all the groups in the table footnote

Discussion: The discussion is insightful and engaging, providing valuable insights into the topic. However, I would suggest considering the inclusion of more data and comparing your findings with additional studies to further strengthen your arguments. This would not only enhance the credibility of your work but also offer a more comprehensive perspective on the subject matter.

Conclusion: The conclusion is well-reasoned and supported by the data presented in the study. However, to make your conclusion even more impactful, I recommend that you include some suggestions for further studies. By offering potential avenues for future research, you can encourage others in the field to build upon your work and explore new dimensions of the topic. This addition would undoubtedly strengthen the significance of your findings and contribute to the advancement of knowledge in the area. Please indicate the key finding about weight gain and FCR.

References: Please provide DOI for all the references.

Although, there are some linguistic errors that should be revised by a language editor. 

Author Response

Dear Reviewer,

Thank you so much for a thorough examination of our MS. We hope that all your criticisms were taken I account properly. Please, find revised version. Revisions introduced following your criticism are highlighted in yellow.

On behalf of the author team,

Sincerely Elena

In the abstract grouping is not clear. Authors should indicate how many groups enrolled in this study, and the numbers and ages of animals. In addition, authors should indicate the duration of the study and introduce sampling parameters in the abstract. The authors mentioned that data about the environmental factors and seasons remain insufficient but they did not provide any data in the conclusion about this question; Therefore, it is recommended that changing the paragraph with other data or at the conclusion it is essential to provide some answer to the question.

Please, consult the revised version of the abstract.

Minor comments:

Line 15: Authors should provide species of grass.

Done

Line 16: Authors should provide full words for NGS

We used a term ‘metagenomic sequencing’ instead of NGS along the whole text.

Line 17: Authors mentioned: 1% of the grass meal. Do the authors mean 1% of dry matter in the diet?

We rephrased this sentence as following ‘Four experimental groups were fed on the diet supplemented with 1% of grass meal over the period 7-42 days of life, no commercial medicines were used here.’

Line 18: Please indicate the route of administration of medicines (toltrazuril and Enrostin)

We rephrased this sentence as following ‘The positive control group (PC) obtained 16 - 25 mg/ml toltrazuril (coccidiostatic agent) and 0.5 ml/L liquid antibiotic Enrostin (100 mg/ml ciprofloxacin and 106 MU/ml colistin sulfate in the commercial preparation) within the drinking water, while the negative control group (NC) obtained no medicines.’

Line 19: Please indicate the dosage of Enrostin (antibiotic)

See previous paragraph.

Line 23: Authors mentioned that: grass meal may substitute the antibiotic for protection against parasitic invasion. But antibiotics are commonly used against bacteria, not parasite infection.

We rephrased this sentence as following ‘Taken together, these observations allow hypothesis that the grass meal may substitute toltrazuril for protecting the chickens from parasitic invasion and increases average daily weight gain (ADG) as effectively as the antibiotic Enrostin.’

Introduction:

Major comments:

The introduction is very long with a huge explanation and data; therefore, authors should minimize the introduction to 1000-1500 words. There are many linguistic and punctuation errors in the introduction.

About 50% text is taken from the Introduction and placed to Discussion section.

Minor comments:

Line 29: Authors mentioned that: Poultry farming is the most efficient meat-producing system. It is not clear that they compare the meat-producing system in poultry farming to what system?

We quoted this sentence directly from the cited paper. However, we somewhat rephrased is as following ‘Now poultry farming is the principal source of meat worldwide, providing the most available source of valuable protein [1].’

Line 34: please change “42nd day of life” to 42 days of life

We changed this syntax along the whole text.

Line 35 and 36: please rephrase

Done

Line 39-41: Please provide a reference for the statement “Antibiotics are commonly used at low doses for infectious disease prevention in broilers, thereby ameliorating their growth and preventing losses”.

Reference [2] is added: Chen J, Wang P, Liu C, Yin Q, Chang J, Wang L, Jin S, Zhou T, Zhu Q, Lu F. Effects of compound feed additive on growth performance and intestinal microbiota of broilers. Poult Sci. 2023 Jan;102(1):102302. doi: 10.1016/j.psj.2022.102302. Epub 2022 Oct 29.

Line 47: Please provide the reference for the statement “global consumption of antimicrobials was expected to be increased by 67%, from 63,151 ± 1,560 tons to 105,596 ± 3,605 tons between 2010 and 2023.

Reference [14] is quoted again (it has been quoted in the previous sentence).

Line 54: As a suggestion you can use the following reference:

Mirzaei, A., Razavi , S. A., Babazadeh, D., Laven, R., & Saeed, M. (2022). Roles of Probiotics in Farm Animals: A Review. Farm Animal Health and Nutrition, 1(1), 17–25. https://doi.org/10.58803/fahn.v1i1.8

Done

Line 87: in the statement “impact of Following preparations “the following should start with a lowercase letter.

Done

Lines 87-90: The paragraph should be rephrased.

We rephrased this sentence as following ‘Therefore, the mechanism and final result of anti-pathogenic action of the antibiotics and probiotics may be more similar than it was suggested formerly.’

Methods and materials:

Major comments:

In the materials and methods authors did not indicate how many replications they had in this trial. The authors should provide the chemical composition of the starter diet and grass meal. Authors should indicate the parameters that they measured in this study in the Chickens and biological trials section. The Mann–Whitney U test is a nonparametric test to compare outcomes between two independent groups. This study contains more than two groups. What test authors used to evaluate the significance between the groups of parameters such as weight gain in the study?

We described the number of the birds thoroughly. There were no more replications in experiments described in MS. The chemical composition of the commercial starter diet PK-5-1is not declared by the manufacturer, but it is popular at commercial poultry farms in Russia. We don’t understand well what do you mean as a chemical composition of the grass meal? We would appreciate your comments about this point.

In the materials and methods authors should provide data that how animals were infected to Eimeria.

Eimeria invasion in our experiment happened due a spontaneous onset, not due to an experimental challenge. We tried to specify this fact in the revised version of the Introduction.

Minor comments:

Line 210-211: Please provide a reference for primers

Done

Line 249: Please indicate the database and move the web address to the reference section.

We don’t like to introduce web address to the reference list since it is often composed automatically on the basis of DOI. In this case we will create a problem for the technical editors. Therefore, we just reduced the URL spelling in the text.

Line 257: Please indicate the country of the academy

Done

Line 258: Please indicate the size of the cage.

Done

Line 258: The authors indicate that “The chickens had ad libitum access to water”. what about feed?

The items of feeding the birds are extensively described in the next paragraph. We don’t se need to repeat this.

Line 264: Please provide a reference for the method of allocating the animals (pairs of analogues)

Our previous work ‘Isakova EP, Serdyuk EG, Gessler NN, Trubnikova EV, Biryukova YK, Epova EY, Deryabina YI, Nikolaev AV. A New Recombinant Strain of Yarrowia lipolytica Producing Encapsulated Phytase from Obesumbacterium proteus. Dokl Biochem Biophys.; vol. 481, no. 1, pp. 201-204, 2018.’ is quoted as a reference [47].

Line 271: Please indicate the exact time of providing a diet to chickens.

We specified ‘(about 8 AM and 6 PM)’

Lines 281-285: Please provide the statistical analysis in a separate subsection.

Section ‘2.4. Statistical analysis’ is added.

Line 282: The Mann–Whitney U test is a nonparametric test to compare outcomes between two independent groups. This study contains more than two groups. What test authors used to evaluate the significance between the groups of parameters such as weight gain in the study?

We used the Mann–Whitney U test for comparison of the experimental groups growth performance parameters to the NC (a) and PC (b) groups. This is specified in a capture to the Table 4.

Line 288: Please provide a reference for the method of sacrificing the animals.

Our previous work ‘Isakova EP, Serdyuk EG, Gessler NN, Trubnikova EV, Biryukova YK, Epova EY, Deryabina YI, Nikolaev AV. A New Recombinant Strain of Yarrowia lipolytica Producing Encapsulated Phytase from Obesumbacterium proteus. Dokl Biochem Biophys.; vol. 481, no. 1, pp. 201-204, 2018.’ is quoted as a reference [47].

Reviewer 3 Report

Line-13: Do not start sentence with But. Instead use However

Line-19: Replace 7 with Seven

Line-19-20: 7 chickens from 19 19 died from coccidiosis in a period of 7-14 days of life in NC group. Two chickens died in the two 20 experimental groups. No losses were registered in other groups (two experimental and PC)??? Reconsider the statement and make it clear.

Line 29-34: paragraph should be excluded.

Line 36-54: Conscise the paragraph..........................

The manuscript needs to be revised. Please make it concise and easy to understandable.

Needs improvement. Make the sentences short and easy.

Author Response

Dear Reviewer,

Thank you so much for a thorough examination of our MS. We hope that all your criticisms were taken I account properly. Please, find revised version.

On behalf of the author team,

Sincerely Elena

Comments and Suggestions for Authors

Line-13: Do not start sentence with But. Instead use However

Done

Line-19: Replace 7 with Seven

The abstract has been changed and the criticism became not actual.

Line-19-20: 7 chickens from 19 19 died from coccidiosis in a period of 7-14 days of life in NC group. Two chickens died in the two 20 experimental groups. No losses were registered in other groups (two experimental and PC)??? Reconsider the statement and make it clear.

Please, see the revised version of the abstract. We hope that it became more understandable.

Line 29-34: paragraph should be excluded.

We suggest that this paragraph is required to explain necessity to substitute antibiotics in the poultry farming.

Line 36-54: Conscise the paragraph..........................

This paragraph was revised following criticism by Reviewer 2.

The manuscript needs to be revised. Please make it concise and easy to understandable.

We hope that the revisions carried out following instructions of all reviewers allowed making the manuscript more comprehensive.

Reviewer 4 Report

Comments for authors

 The article “Grass meal acts as a probiotic in chicken” covers a much-needed area of research, i.e. need for new additives that can be used as growth promoters and has the ability to replace antibiotics. Although the test product “grass biomass” seems to be able to promote growth, the article is rejected in its current form as it is poorly designed, and all sections need to be rewritten to make the content and tables clear and precise. A few major comments below may help to improve the article.

 Abstract: 7 out of 19 birds died in the NC group without coccidiostats, which suggests that the mortality level was around 36%. This is a welfare concern. Secondly, parasitic invasion was not checked in birds in other treatment groups; therefore, the conclusion “grass meal may substitute the antibiotic for protection against parasitic invasion” is not justified. This statement was only justified if a representative number of birds from each treatment group were dissected to check for Emeria lesion scores or faecal samples were to be taken for Emeria spp. count. Therefore, instead of claiming that the study proves to prevent parasitic invasion, you can say the study showed improved growth.

 Introduction: Too long. It should not be more than one and a half pages. Some of the parts can be moved to the discussion section.

Line 183: “infected with Eimeria” How?

 Material and Methods:

This section needs to be split into sections:  

·         Collection and drying of grass specimens,

·         In-vitro testing of grass specimens:

o    Composition

o   Bacterial contamination

·         In-vivo trial

o   Birds and housing: what type of cages? What was the stocking density during the trial period? Were the battery cages wired, or did they have floor pens?

o   Experimental diets and test additive dosage: How was grass biomass added to the diet and fed? Not clear. Why was grass biomass offered daily rather than incorporated into the diet formulation and part of the diet? Be consistent with your use of terminology. Is this grass biomass or feed additives? Not appropriate to call it fodder or hay.

o   Pretrial period: Why were all birds given antibiotics in the pretrial phase? By introducing antibiotics, the natural gut microbiota of birds is disturbed. Therefore difficult to confirm that the positive effect is all due to grass addtive? What is PK-5-1?

o   How were birds distributed in cages? Was it a completely randomised block design?

o   Emeria challenge model adopted?

o   Experimental period and parameters measured: Must mention how many samples/treatment were collected, when and which analysis was conducted.

o   Microbiological Analysis

o   Statistical analysis

Results and discussion: Results section has a mix of results and discussion. The discussion part should be removed. This part should not have references.

Mortality data should be presented in percentages. What was the overall mortality?

 Were FCR values corrected for mortality?

 Line 420: how was grass heated?

Line 455: no death does not mean the grass biomass prevented mortality. Do you have lesion scoring or quantitive data confirming that other birds were healthy?

Line 475: what do you mean by protection from invasion and infection?

 Table 2 is not required as it does not add any additional information.

 Tables 4 and 5 can be merged and provide complete performance add, including body weight, weight gain, feed intake and FCR for all periods plus, it will be beneficial to include the overall period days 7 to 42. The table should have a column representing SED and P-value.

 Line 501: “This additive completely prevented chicken’s death”. How can you provide that?

Line 517: Add feed intake treatment means in the performance table.

 Discussion should explain what is in the grass biomass that gives it the ability to act as a probiotic. Try and explain the possible mode of action.

 The data presented shows that grass biomass improves growth, but it's not robust enough to claim a reduction in parasitic invasion.

 Too many abbreviations. All abbreviations should be used as full words, followed by abbreviations in brackets.

Minor editing of the English language is required.

Author Response

Dear Reviewer,

Thank you so much for a thorough examination of our MS and for the acknowledgement of its overall actuality. We hope that all your criticisms were taken I account properly. Please, find revised version. Revisions introduced following your criticism are highlighted in magenta.

On behalf of the author team,

Sincerely Elena

The article “Grass meal acts as a probiotic in chicken” covers a much-needed area of research, i.e. need for new additives that can be used as growth promoters and has the ability to replace antibiotics. Although the test product “grass biomass” seems to be able to promote growth, the article is rejected in its current form as it is poorly designed, and all sections need to be rewritten to make the content and tables clear and precise. A few major comments below may help to improve the article.

Abstract: 7 out of 19 birds died in the NC group without coccidiostats, which suggests that the mortality level was around 36%. This is a welfare concern. Secondly, parasitic invasion was not checked in birds in other treatment groups; therefore, the conclusion “grass meal may substitute the antibiotic for protection against parasitic invasion” is not justified. This statement was only justified if a representative number of birds from each treatment group were dissected to check for Emeria lesion scores or faecal samples were to be taken for Emeria spp. count. Therefore, instead of claiming that the study proves to prevent parasitic invasion, you can say the study showed improved growth.

We don’t claim that the birds in the grouped beyond NC were not infected. Most likely, they encountered a parasite and suffered an invasion. However, in contract to NC group they survived until the end of experiment and exhibited high growth performance parameters which is impossible in the birds that sick from invasion. In our opinion this proves that the grass meal conferred a resistance on the birds.

Introduction: Too long. It should not be more than one and a half pages. Some of the parts can be moved to the discussion section.

We moved about a half of the introduction text to Discussion.

Line 183: “infected with Eimeria” How?

The invasion of our birds in the course of the experiment was spontaneous. No challenge was carried out. Perhaps, we must specify that there was an extensive outbreak of coccidiois in Krasnodar region of Russia in summer 2022. We introduced the following paragraph to Materials and Methods for explaining our approach to Eimeria invasion diagnosis ‘The DNA samples isolated from the ileum digesta were analyzed by PCR with primers EtF (AATTTAGTCCATCGCAACCCT) and EtR (CGAGCGCTCTGCATACGACA) specific to ITS-1 of the ribosome cluster [48] and sequenced by Sanger method by using BigDye Terminator v3.1 Cycle Sequencing Kit (Thermo Fisher Scientific, USA) and Nanophore 05 genetic analyzer (Syntol, Russia) once PCR products appeared. The derived sequences were compared to NCBI GenBank. Their affiliation to the genomic DNA of Eimeria tenella was verified by similarity with Eimeria tenella genome assembly, chromosome 13 (NCBI GenBank Accession number HG994973).’.

 Material and Methods:

This section needs to be split into sections: 

  • Collection and drying of grass specimens

Done

  • In-vitro testing of grass specimens:

Done

o    Composition

o   Bacterial contamination

  • In-vivo trial

o   Birds and housing: what type of cages? What was the stocking density during the trial period? Were the battery cages wired, or did they have floor pens?

We described this as following: ‘First 7 days 130 one-day old Ross 308 cross broilers were placed to vivarium of the Skryabin Academy of Veterinary Medicine and Biotechnology (Moscow, Russia) and kept at temperature 32±1°C on a 12-h photoperiod in cage batteries with a mesh floor with an area of 80×90 cm, 20 heads per a cage’. And later: ‘All birds were kept in cages with a concrete floor with an area 1.5×2 m covered with a sawdust litter which was changes twice a week.’.

o   Experimental diets and test additive dosage: How was grass biomass added to the diet and fed? Not clear. Why was grass biomass offered daily rather than incorporated into the diet formulation and part of the diet? Be consistent with your use of terminology. Is this grass biomass or feed additives? Not appropriate to call it fodder or hay.

We extended the respective paragraph as following: ‘Further the experiment was carried out until 42 day of life (35 days). This period the chickens were kept on a floor. Each group had ad libitum access to the food. The complete diet without antibiotics Ekorm-ROST grower diet purchased from Stavropolsy kombikorm (Russia) was provided in excess twice a day about 8 AM and 6 PM, each diet portion was weighted. Each experimental diet was prepared for the whole period of experiment by adding 1% of the respective grass meal sample to the whole volume of the diet (2 kg of the grass meal per 200 kg Ekorm-ROST diet) and mixing in 100 L hopper with propeller stir-rer EuroPlast (Russia). Enrostin and toltrazuril were not mixed with the diet since the were administrated to PC group birds with a drinking water’.

o   Pretrial period: Why were all birds given antibiotics in the pretrial phase? By introducing antibiotics, the natural gut microbiota of birds is disturbed. Therefore difficult to confirm that the positive effect is all due to grass addtive? What is PK-5-1?

We can’t specify exact composition of PK-5-1 starter diet, but we know that is popular at industrial poultry farms in Russia. When concerning antibiotics, you can see our experimental data that confirm that Enrostin antibiotic together with toltrazuril substantially promote growth performance parameters of the birds when PC group is compared to NC group. This clearly explain why it is used at most poultry farms in Russia (I don’t know exactly what do in practice in other countries). We used antibiotic in the pretrial period since it is prerequisite for the flock survival. This was carried out uniformly in all groups. Therefore, in our opinion, comparison of the mortality and growth performance parameters between the groups looks correct.

o   How were birds distributed in cages? Was it a completely randomised block design?

We added a reference for our previous work describing this principle. Besides, data about evenness of the initial ABW in groups are shown in Table 2. In our opinion, they give an evidence an appropriate equilibration of ABW between the groups indeed.

o   Emeria challenge model adopted?

We didn’t use experimental challenge of the birds with Eimeria. We specified in Introduction and Results that this was just a spontaneous invasion.

o   Experimental period and parameters measured: Must mention how many samples/treatment were collected, when and which analysis was conducted.

I don’t understand this question. Please, explain in more detail.

o   Microbiological Analysis

o   Statistical analysis

Results and discussion: Results section has a mix of results and discussion. The discussion part should be removed. This part should not have references.

All references in this section are given in order to illustrate what kind of bacteria were found by metagenomic sequencing and microbiological seeding from the grass specimens. There is no appropriate place for this in Discussion. I suggest to leave description of the grass microbiome in Results since otherwise it will be too difficult to trace the logics in Discussion.

Mortality data should be presented in percentages. What was the overall mortality?

We specified mortality rate in absolute figures and in %: NC group - 7 from 20 is 35%, KS2 – 2 from 20 is 10%; TS1 – 3 from 20 is 15%.

Were FCR values corrected for mortality?

We described algorithm of FCR calculation in Statistical analysis section.

Line 420: how was grass heated?

We have the following paragraph in Materials and Methods section: ‘The count of Bacilli sensu lato (number of live thermostable endospores) was deter-mined as following: 50 l samples of the dry grass meal were placed into 1.5 ml Eppendorf tube, 1.000 l sterile deionized water added and incubated at 90°C for 10 min without preliminary mixing. Then the samples were thoroughly mixed at a hand vortex and 100-times diluted by transferring 10 l aliquots of the heated grass extract into 1.000 l volumes of sterile deionized water in new tubes. 10 ml aliquots of each heated grass ex-tract and its 100-folds dilution were distributed onto LB agar plates (pepton bacto 10 g/L, yeast extract bacto 5 g/L, NaCl 5 g/L, agar bacto 15 g/L). Each specimen was analyzed in duplicate. The plates were incubated at 30°C for 48 hours and number of colonies was calculated manually. The number of colonies in range 20-200 per plate was suggested to be adequate for accurate calculation of the initial contamination of the grass sample with the endospore-forming bacteria.’. Besides we added the following paragraph to Results section: ‘For isolating Bacilli sensu lato, 50 mg grinded grass samples were placed to 1.7 ml eppendorf tubes, flooded with 1 ml deionized water and heated for 10 min in solid-state thermostat for tubes. The tubes were not mixed before heating to avoid casting the bacteria containing material to the not-heated lid. After the heating, the samples were thoroughly mixed by vortexation, 50 µl aliquots of the suspension were picked up by a trimmed 200 µl automated pipet tip, placed onto the nutrient agar and distributed by a glass spatula. For isolating Proteobacteria, the grinded grass suspensions prepared by a similar way without heating were picked up by a trimmed 200 µl automated pipet tip, placed onto the nutrient agar containing 35 g/l erythromycin and distributed by a glass spatula. The respective experimental procedures are described in detail in section 2.1’.

Line 455: no death does not mean the grass biomass prevented mortality. Do you have lesion scoring or quantitive data confirming that other birds were healthy?

We suggest that obviously good growth performance parameters in the experimental group indistinguishable from PC and confidentially different from NC give an evidence that Eimeria invasion was not dangerous for the birds although they obviously faced the parasite.

Line 475: what do you mean by protection from invasion and infection?

We mean that all birds in PC, KS1 and TS2 groups were safe and that they have optimal growth performance parameters in contrast to alive birds from NC group.

Table 2 is not required as it does not add any additional information.

We suggest that is necessary in order to prove even distribution of the birds into the groups by using method of pairs of analogues. It will be difficult to explain this without this table.

 Tables 4 and 5 can be merged and provide complete performance add, including body weight, weight gain, feed intake and FCR for all periods plus, it will be beneficial to include the overall period days 7 to 42. The table should have a column representing SED and P-value.

This is done.

Line 501: “This additive completely prevented chicken’s death”. How can you provide that?

There was no mortality in PC, KS1 and TS2 groups in contrast to NC, KS2 and TS1 groups.

Line 517: Add feed intake treatment means in the performance table.

Please, see Table 4

Discussion should explain what is in the grass biomass that gives it the ability to act as a probiotic. Try and explain the possible mode of action.

Please see discussion section. Briefly, a beneficial impact of Bacillli on the GIT microbiota is well documented, and we have a substantial colonization of the grass with this group of bacteria. If TS2 grass sample is considered, there is no substantial colonization with Bacilli and we suggest Enterobacteria to be responsible for the probiotic-like effect.

The data presented shows that grass biomass improves growth, but it's not robust enough to claim a reduction in parasitic invasion.

We suggest that 0 dead birds aren’t the same as 7, are they? Besides, the growth performance parameters in NC group are confidentially wore than in others. This would be impossible under effective proliferation of Eimeria in GIT.

Too many abbreviations. All abbreviations should be used as full words, followed by abbreviations in brackets.

We tried to reduce the number of abbreviations and controlled the order of the first use of each.

Reviewer 5 Report

The author studied the Grass meal's effect on the probiotic in chicken, before consideration some suggestion needs to be revised below.

Revise the title, clearly mention such as effect of grass meal on growth performance or probiotic etc.

Mention the type of chicken in the Abstract (Broiler or layer etc.?)

The introduction is very lengthy, please revise it.

Add the aim of your study at the end of the introduction!

Add separate subheading for 16S DNA analysis from the Plant biomass specimens.

Line 256-259 revises the sentence for English grammar.

Line 281-285 the statistical information should be provided in the separate statistical analysis subheading, delete from this section.

Discussion of the manuscript is very poor and short, but interlocution is very lengthy, author should revise discussion and provide summarize the major gap in understanding that your work is attempting to fill. What was the overarching hypothesis? What was the most important result of your study?

In the introduction an effective way to express the objectives of the study is to combine the problem with what has been done to solve it and present it in a single sentence.

 Extensive editing of English language required for the manuscript

 Extensive editing of English language required

Author Response

Dear Reviewer,

Thank you so much for a thorough examination of our MS. We hope that all your criticisms were taken I account properly. Please, find revised version. Revisions introduced following your criticism are highlighted in green.

On behalf of the author team,

Sincerely Elena

The author studied the Grass meal's effect on the probiotic in chicken, before consideration some suggestion needs to be revised below.

Revise the title, clearly mention such as effect of grass meal on growth performance or probiotic etc.

Done

Mention the type of chicken in the Abstract (Broiler or layer etc.?)

Done

The introduction is very lengthy, please revise it.

Done

Add the aim of your study at the end of the introduction!

Done

Add separate subheading for 16S DNA analysis from the Plant biomass specimens.

Done

Line 256-259 revises the sentence for English grammar.

Done

Line 281-285 the statistical information should be provided in the separate statistical analysis subheading, delete from this section.

Done

Discussion of the manuscript is very poor and short, but interlocution is very lengthy, author should revise discussion and provide summarize the major gap in understanding that your work is attempting to fill. What was the overarching hypothesis? What was the most important result of your study?

We reduced Introduction, gave an extended description of the objective and extended Discussion.

In the introduction an effective way to express the objectives of the study is to combine the problem with what has been done to solve it and present it in a single sentence.

Done

Extensive editing of English language required for the manuscript

We tried to carry out this.

Round 2

Reviewer 2 Report

Thank you very much for improving the quality of the text. 

The article is well-revised and prepared for further processing in the journal. There are just a few issues that can be checked by the language editor before final publication on the journal website.

Author Response

Thank you 

Reviewer 4 Report

Dear authors,

Thanks for addressing some of the comments made by the reviewer. However, a significant amount of time and effort has been spent to re-evaluate the revised manuscript for publication. After careful consideration, it is with regret that the manuscript is still unsuitable for publication in its current form. Below are some of the areas of concern which require further clarification:

·       Introduction section: needs to be rewritten. The comment made previously was not fully taken on board: The introduction should have a paragraph each on the importance of commercial poultry farming, introduce the importance of probiotics in poultry farming, what grass meal is, and what is in it that makes it to act as probiotics, and how it can improve broiler production. Finally, identify the gap in knowledge in this area and the need for your research work and end with what you are trying to achieve from this study (stating study objectives.

·       Methods and material section:

o   Line 290-293: Why PC group has 10 spare birds. Having extra birds in one group means that the stocking density of that group differed from the rest and thus may affect the bird's performance.

o   Line 293 vs 325vs 535: the sentence says ileal digesta was collected for PCR reads and then later you mention caecal and crop? And in the results section it says ileal samples? Which samples were collected? Not clear, rather confusing.  

o   Line 296: Why is sawdust litter changed twice a week?

o   Table 2: This is usual to take representative weights/group : there is no need to proved min and max weights.

o   Line 343: What was the purpose of recording feed intake twice daily

·       Results and discussion: Line 535-537: The claim that only NC birds had parasite invasion is not justified when only dead birds were tested for parasite invasion. This was only possible if an equal number of birds from each group were culled and PCR analysis was conducted on an equal number of samples/treatments.

o   Table 4 : Delete means column but add P-values for each parameter/time point. Superscripts a-b is not clear if they relate to a column or row? Make it clear in the table. It is not clear if feed intake is average feed intake /period or if it is ADFI?

o   The grass biomass genomic data is very good and relevant, which is the novelty of your work. However, the growth performance data is stretched too much. You only have average body weight, which is significantly better than NC and similar to PC. FCR data is numerically different (as no superscripts indicate non-significant data). So you can't mention that as a major difference in FCR (Line 559)   

o   Add a table on PCR parasite invasion data.

o   Line 577: The mention of better digestibility is out of proportion as the feed intake data has no significant difference between the treatment groups.

o   Line 578: Show data to confirm the suppression of growth of E.tenella.

o   Line 581: FCR data description does not match Table 4.

o   Line 631: what does “following preparation” means? Not clear.

Conclusion: it is much improved but still has areas for improvement.

Line 708: The sentence needs clarification.

Line 728: Which data confirms this? Not clear. 

Minor improvement is required.

Author Response

Dear Reviewer, we appreciate your patience, concern about MS quality and particularly serious time expenditures. We apologize that some your criticisms weren’t taken into consideration. This is caused mainly due to necessity to keep in touch with several reviewers in parallel. We are glad to a positive estimate of MS novelty in part of plant microbiome using as a potential probiotic. As far as for Introduction, we don’t see any point mentioned by you which isn’t addressed in this section. - The introduction should have a paragraph each on the importance of commercial poultry farming We have a paragraph ‘Now poultry farming is the principal source of meat worldwide, providing the most available source of valuable protein [1]. An intensive development of four-line cross system in chickens (e.g. Cobb 500 and Ross 308 fast growing bred) and ameliorating cage, ventilation, climatic, feed distributing and waste management facilities over the last 6 or 7 decades has arisen feed stock conversion into muscle mass efficiency [2]. Feed conversion ratio (FCR) in these crosses attains 1.5-1.7 to 42 days of life [3, 4].’ Other Reviewers suggested its complete eliminating, therefore apparently there is no to extend it. Apparently, its present state looks as a compromise between several opinions. - introduce the importance of probiotics in poultry farming, what grass meal is, and what is in it that makes it to act as probiotics, and how it can improve broiler production. This point is the main in the introduction in its present state. We suppose that the addressed question is comprehensively answered by the following speculation: ‘A gross industrial farming of the broilers decreases manufacturing cost but makes the flock vulnerable towards infection with Campylobacter jejuni [5], Clostridium perfringens [6], Clostridium deficile [7], several species of Salmonella [8], enteropathogenic Escherichia coli [9], Eimeria [10], avian leukosis virus [11] and Enterococcus avium, resulting in large economic losses to the poultry industry worldwide [12]. Antibiotics and antiparasitic medicines are commonly used at low doses for infectious disease prevention in broilers, thereby ameliorating their growth and preventing losses. Nevertheless, the misuse and overuse of the drugs as growth promoters unavoidably leads to the emerging antibiotic-resistance in the broiler microbiota including pathogens [13]. In 2015, the global annual consumption of antimicrobials per kg of animal product was estimated as 45 mg/kg, 148 mg/kg, and 172 mg/kg for cattle, chicken, and pigs, respectively [14]. Starting from this baseline, the global consumption of antimicrobials was expected to be increased by 67%, from 63,151 ± 1,560 tons to 105,596 ± 3,605 tons between 2010 and 2030 [14]. The impact of antibiotic use for growth promotion in livestock and poultry production on the rise of antimicrobial resistance in bacteria led to the ban of this practice in the European Union since 2006 and a restriction of antimicrobial use in animal agriculture in Canada and in the US [15]. A recently emerged paradigm of bioeconomy suggests using biological means of control for infection agents affecting poultry, including probiotics (alive microbial preparations with antagonist activity toward pathogens), prebiotics, phytobiotics, bacteriophages and their lysins [16, 17]. Since 1973, the probiotics are suggested as an efficient and safe alternative to the feed antibiotics [2, 18]. Traditionally representatives of genus Lactobacillus and taxonomically close groups (Streptococci, Enterococci) were used in this role [19]. As far as Lactobacilli are normal component of the chicken crop, small intestine and cloaca, but not caeca, a long-term survival of the administrated bacteria was presumed. Therefore, strains with a high resistance to acidic pH, bile and pepsin with high adhesion ability to intestinal mu-cin were suggested to be most efficient and special methods of selection for these traits were proposed [20]. Further probiotics of other taxonomic lineages were successfully used. First, Enterobacteria including E. coli [21] and Bacilli [22] were used. There are reports about high efficiency of B. subtilis strain isolated from chicken feces [23] despite of the fact that survival of Bacillus in the chicken gastric intestinal tract (GIT) seems doubtful. Later strains of Clostridium [24] and Ascomycetes yeast e.g. Saccharomyces boulardii [25] were introduced to practice. The most popular commercial probiotics available on the global market are Aviguard, Primalac and Interbac made up of several species of Lactobacillus and Bacillus [26]. Significantly, probiotics are like antibiotics (e.g. enramycin and tylosin) confer a resistance against Eimeria on the chicken although they do not exhibit antagonism towards Apikomplexa sporozoits in vitro [27]. Moreover, a concept of necessity of a long-term persistence of the probiotics in GIT was revised. Obviously, a high efficiency of Bacillus strains anti-pathogenic and growth-promoting effect on the chicken was acknowledged although a pure aerobic metabolism does not allow Bacilli vegetate under the chicken GIT anaerobic conditions. Moreover, culture medium fermented with Bacillus licheniformis and Bacillus subtilis exerted a favorable impact on the GIT microbiota and average daily weight gain (ADG) of the broilers [2, 28]. This confirms that a short-term influence of the probiotic derived metabolite is sufficient for the favorable action of the overall probiotic. Therefore, the mechanism and final result of anti-pathogenic action of the antibiotics and probiotics may be more similar than it was suggested formerly. Effect of antibiotics and probiotics of GIT microbiota in chicken were extensively studied by using metagenome sequencing (amplified libraries of 16S rDNA gene fragments were sequenced on Illumina platform) [29, 30, 31]. It has been indicated that in the caeca of broilers Clostridia are the predominate organisms [32], while genus Lactobacillus is dominated in the ileum [33].’ - Finally, identify the gap in knowledge in this area and the need for your research work and end with what you are trying to achieve from this study (stating study objectives. No we have a firmly formulated Objective of the study: ‘Taking into consideration this idea, the present study had an objective to estimate growth-stimulating effects of the grass meal, specimens collected in two distinct geographic locations on the chicken in comparison to a negative control group (obtaining no medicines) and a positive control group fed a diet supplemented with a feeding antibiotic Enrostin (100 mg/ml ciprofloxacin and 106 MU/ml colistin sulfate). Bacterial load of the grass meal specimens was qualitatively and quantitatively assessed by using meta-genomic sequencing of libraries obtained with Ferier_F515 and Ferier_R806 primers spe-cific to V4 region of 16S ribosomal DNA [43]. A particular attention was paid to endospore-forming bacteria (Bacilli sensu lato) which are relatively wide spread in phylloplane [44] and have been suggested as efficient veterinary probiotics [2, 28]. In parallel, a protective effect of the grass meal specimens towards spontaneous invasion of the chickens with Eimeria tenella (coccidiosis) was studied. Molecular analysis by PCR with primers EtF and EtR specific to ITS-1 of the ribosome cluster [45] demonstrated presence of this parasite in the ileum digesta of 6 chickens from 7 dead in the negative control group in the course of the trials in the single dead chicken from experimental group KS2. One more chicken from NC group, a chicken from KS2 experimental group and three chickens from TS1 experimental group which died in the course of the trials as well as in none chickens survived until the end of the trials exhibited no E. tenella DNA in the ileum digesta. This observation allows hypothesizing that the grass meal may confer a specific anti-coccidiosis effect on the chicken or exhibits an overall restorative effect, increasing their resistance to parasitic invasions’. We apologize for our inability to take on a board your concern since we see no gaps in the text composition in the mentioned points. Dear authors, Thanks for addressing some of the comments made by the reviewer. However, a significant amount of time and effort has been spent to re-evaluate the revised manuscript for publication. After careful consideration, it is with regret that the manuscript is still unsuitable for publication in its current form. Below are some of the areas of concern which require further clarification: Introduction section: needs to be rewritten. The comment made previously was not fully taken on board: The introduction should have a paragraph each on the importance of commercial poultry farming, introduce the importance of probiotics in poultry farming, what grass meal is, and what is in it that makes it to act as probiotics, and how it can improve broiler production. Finally, identify the gap in knowledge in this area and the need for your research work and end with what you are trying to achieve from this study (stating study objectives. · Methods and material section: o Line 290-293: Why PC group has 10 spare birds. Having extra birds in one group means that the stocking density of that group differed from the rest and thus may affect the bird's performance. We extended the paragraph as following to avoid confusion: ‘Ten birds not included to the experimental groups were kept in a separate cage as a reserve on PC group diet containing toltrazuril and Enrostin. They were not taken into account when the growth performance parameters were assessed and were used as a negative control for E. tenella PCR diagnosis”. o Line 293 vs 325vs 535: the sentence says ileal digesta was collected for PCR reads and then later you mention caecal and crop? And in the Results section it says ileal samples? Which samples were collected? Not clear, rather confusing. Thank you for your attention! No samples from the crops and caeca were analyzed. We corrected the text in Results section as following: ‘The dead chickens were subjected to autopsy for collecting ileal digesta specimens which were immediately frozen. The control healthy chickens from the reserve group were humanely killed through carbon dioxide inhalation in the same age while spontaneous death due to contamination was registered in the negative-control group. 100 mg ileal digesta were sampled in duplicate from each chick and used for DNA purification by using K-sorb micro-column sorbent kit (Syntol, Moscow, Russia) following instructions by manufacturer’. o Line 296: Why is sawdust litter changed twice a week? This is a standard practice in our vivarium. Why is this of interest? o Table 2: This is usual to take representative weights/group: there is no need to proved min and max weights. We would like to keep min and max values in Table 2 to let see to the reader extent of evenness of our experimental groups. As far as we have relatively small groups, an abnormal initial weight of even one bird can distort the results. In our opinion, the data in Table 2 ensure that there was no inaccuracy in our experiment in this point. You have asked above about the reason to exclude 10 birds from the experiment. You can see here that this is a cost of aligning the experimental groups. o Line 343: What was the purpose of recording feed intake twice daily The diet was given to the birds twice a day in excess to make them free access to the food. We suggested that therefore we minimize risk of the diet spoilage. If so, there was no another way to measure the weight of the remaining/eaten food: we must weight the rest of the previous portion just before adding the fresh one. Results and discussion: Line 535-537: The claim that only NC birds had parasite invasion is not justified when only dead birds were tested for parasite invasion. This was only possible if an equal number of birds from each group were culled and PCR analysis was conducted on an equal number of samples/treatments. As mentioned above, we had only 10 heads in the reserve group, and this is lesser than in the experimental and control groups. All these birds were killed in the same period (days 14-21) when all losses were registered in NC, KS2 and TS1 groups. we found no parasite in the ileum digesta of the reserve group. This is reported in the text and in the new added Table 4: ‘The PCR positive in the samples from 6 chickens from 7 died in the NC group and in the single dead chicken from experimental group KS2 (Table 4). The PCR products were sequenced by Sanger method. The derived sequences were compared to NCBI GenBank and exhibited 100% similarity with Eimeria tenella genome assembly, chromosome 13 (NCBI GenBank Accession number HG994973). Taken together, these data unambiguously proves that the death from coccidiosis (spontaneous invasion with E. tenella) in the NC group attained 37% and these losses were completely prevented by either combination of toltrazuril and Enrostin (positive control group) or by grass meal (experimental groups KS1, TS1, TS2) and partially prevented in KS2 group. No PCR products with primers EtF and EtR were found in the ileum digesta DNA of the reserve group (totally 10 heads) which were sacrificed simultaneously with the birds died from the invasion’. o Table 4: Delete means column but add P-values for each parameter/time point. Superscripts a-b is not clear if they relate to a column or row? Make it clear in the table. It is not clear if feed intake is average feed intake /period or if it is ADFI? We don’t agree that the only P-values provide a relevant perception of the growth performance parameters, the absolute figures are of interest as well since they allow a reader to have an idea concerning overall ‘life quality of the flock’. Absolute values allow comparison of our results to others reported elsewhere. Therefore, we would like to propose to retain the table in its present state. We extended description of a and b superscript indexes in the table legend: a - statistically confident difference in the parameter value in comparison to the positive control group to the same time point of the experiment (p

Reviewer 5 Report

The Subheading of Statistical analysis should be after the 16S DNA library construction, sequencing and bioinformatics analysis subheading. The statistical analysis information should be always last part of the methodology.

minor revisions

Author Response

Dear Reviewer,

Thank you for your efforts, spent time and for the consent to give a report.

Best regards,

Sincerely yours Elena